# Development of antibody-dependent cell cytotoxicity function in HIV-1 antibodies

**Laura E Doepker[1†], Sonja Danon[1†], Elias Harkins[2], Duncan K Ralph[2], Zak Yaffe[1,3], Meghan E Garrett[1,4], Amrit Dhar[2,5], Cassia Wagner[3], Megan M Stumpf[1], Dana Arenz[1], James A Williams[6], Walter Jaoko[7], Kishor Mandaliya[8], Kelly K Lee[6], Frederick A Matsen IV[2], Julie M Overbaugh[1,2]\***

[1]Human Biology Division, Fred Hutchinson Cancer Research Center, Seattle, United States; [2]Public Health Sciences Division, Fred Hutchinson Cancer Research Center, Seattle, United States; [3]Medical Scientist Training Program, University of Washington School of Medicine, Seattle , United States; [4]Molecular and Cellular Biology Graduate Program, University of Washington and Fred Hutchinson Cancer Research Center, Seattle, United States; [5]Department of Statistics, University of Washington, Seattle, United States; [6]Department of Medicinal Chemistry, University of Washington, Seattle, United States; [7]Department of Medicinal Microbiology, University of Nairobi, Nairobi, Kenya; [8]Coast Provincial General Hospital, Women's Health Project, Mombasa, Kenya

**\*For correspondence:**
joverbau@fredhutch.org

[†]These authors contributed equally to this work

**Abstract** A prerequisite for the design of an HIV vaccine that elicits protective antibodies is understanding the developmental pathways that result in desirable antibody features. The development of antibodies that mediate antibody-dependent cellular cytotoxicity (ADCC) is particularly relevant because such antibodies have been associated with HIV protection in humans. We reconstructed the developmental pathways of six human HIV-specific ADCC antibodies using longitudinal antibody sequencing data. Most of the inferred naive antibodies did not mediate detectable ADCC. Gain of antigen binding and ADCC function typically required mutations in complementarity determining regions of one or both chains. Enhancement of ADCC potency often required additional mutations in framework regions. Antigen binding affinity and ADCC activity were correlated, but affinity alone was not sufficient to predict ADCC potency. Thus, elicitation of broadly active ADCC antibodies may require mutations that enable high-affinity antigen recognition along with mutations that optimize factors contributing to functional ADCC activity.

## Introduction

A major concept underlying HIV vaccine research is that we can learn from the processes that lead to potent antibody responses in natural infection to inform immunogen design. For this reason, there have been several detailed studies of the evolutionary pathways of potent and broad neutralizing antibodies (bnAbs) to HIV (*Bonsignori et al., 2017*; *Bonsignori et al., 2016*; *Doria-Rose et al., 2014*; *Landais et al., 2017*; *Liao et al., 2013*; *MacLeod et al., 2016*; *Simonich et al., 2019*; *Wu et al., 2015*). These studies highlight the critical role of somatic hypermutation (SHM), particularly in complementary determining regions (CDRs) of antibody heavy chains, in driving the breadth and potency of HIV bnAbs (*Kwong and Mascola, 2018*). One recent study also highlighted that maturation of the antibody light chain, too, can be critical for breadth (*Simonich et al., 2019*). Collectively, these studies have provided important insights into the specific mutations that drive maturation of antibody lineages and specificities of bnAbs.

**eLife digest** Nearly four decades after the human immunodeficiency virus (HIV for short) was first identified, the search for a vaccine still continues. An effective immunisation would require elements that coax the human immune system into making HIV-specific antibodies – the proteins that can recognise, bind to and deactivate the virus.

Crucially, antibodies can also help white blood cells to target and destroy cells infected with HIV. This 'antibody-dependent cellular cytotoxicity' could be a key element of a successful vaccine, yet it has received less attention than the ability for antibodies to directly neutralize the virus. In particular, it is still unclear how antibodies develop the ability to flag HIV-infected cells for killing. Indeed, over the course of an HIV infection, an immune cell goes through genetic changes that tweak the 3D structure of the antibodies it manufactures. This process can improve the antibodies' ability to fight off the virus, but it was still unclear how it would shape antibody-dependent cellular cytotoxicity.

To investigate this question, Doepker et al. retraced how the genes coding for six antibody families changed over time in an HIV-carrying individual. This revealed that antibodies could not initially trigger antibody-dependent cellular cytotoxicity. The property emerged and improved thanks to two types of alterations in the genetic sequences. One set of changes increased how tightly the antibodies could bind to the virus, targeting sections of the antibodies that can often vary. The second set likely altered the 3D structure in others ways, potentially affecting how antibodies bind the virus or how they interact with components of the immune system that help to kill HIV-infected cells. These alterations took place in segments of the antibodies that undergo less change over time. Ultimately, the findings by Doepker et al. suggest that an efficient HIV vaccine may rely on helping antibodies to evolve so they can bind more tightly to the virus and trigger cellular cytotoxicity more strongly.

The keen interest in HIV bnAbs stems in part from a plethora of compelling studies that demonstrate the efficacy of bnAbs in protection of macaques from experimental SHIV infection (*Pegu et al., 2017*). The data are less convincing for sterilizing protection by HIV-specific non-neutralizing antibodies that mediate antibody-dependent cell cytotoxicity (ADCC) in the same experimental animal models (*Fouts et al., 2015*), where the functional interactions of antibodies are harder to test due to species differences (*Bournazos et al., 2017*). However, they have been implicated in protection from HIV-infection in humans in several settings. In the only partially effective HIV vaccine trial to date, antibodies that mediate ADCC were associated with protection, whereas neutralizing antibodies were not (*Haynes et al., 2012*). Moreover, in the setting of mother-to-child transmission, ADCC antibodies have been associated with both decreased transmission risk and disease progression in infants (*Mabuka et al., 2012*; *Milligan et al., 2015*). For these reasons, ADCC antibodies, which have the potential to eliminate infected cells, have been increasingly recognized as important to consider for vaccine design and prevention efforts (*Lewis et al., 2017*). Yet, nothing is known about the process of SHM required for HIV specificity and ADCC function of antibodies.

In light of the relevance of ADCC antibodies in human infections, we deep sequenced the antibody repertoire of an individual who developed potent ADCC antibodies to the HIV envelope (Env) gp120 and gp41 proteins. We used computational methods that were specifically developed to study antibody evolutionary pathways to infer the naïve precursors and resolve chronological lineage intermediates leading to six different mature ADCC-mediating IgG1 antibodies that have been previously described (*Williams et al., 2015*; *Williams et al., 2019*). Among the six lineages, most inferred naïve antibodies did not mediate ADCC. To develop ADCC functionality, inferred naive antibodies required CDR-localized SHM in one or both chains. In five of six cases, enhancement of ADCC potency to mature levels required mutations in framework regions (FWR), either alone or alongside additional CDR mutations. Interestingly, in one lineage, the developing antibodies first gained the capacity to bind HIV Env and then subsequently acquired ADCC activity due to additional SHM. Overall, binding affinity for Env and facilitation of ADCC were correlated, but we observed cases where binding affinity was similar amongst two antibodies within the same lineage but they differed in their ADCC capacity, suggesting that affinity alone is not sufficient for an antibody to mediate ADCC.

**Table 1.** Longitudinal QA255 antibody variable region sequencing statistics.

| Time point | Live PBMC count (excluding non-viable) | PBMC viability | Ab chain | Replicate | Raw MiSeq reads | Productive replicate-merged deduplicated sequences | Estimated sequencing coverage within sampled blood PBMCs[†] | Estimated sequencing coverage within QA255 d914 whole-body blood repertoire[‡] |
|---|---|---|---|---|---|---|---|---|
| D-119 | 1.39E+07 | 84.22% | IgM | 1 | 486,458 | 200,254 | 21% | 0.05% |
| | | | IgG | 1 | 604,237 | 199,570 | 48% | 0.11% |
| | | | IgK | 1 | 705,163 | 292,543 | 21% | 0.05% |
| | | | IgL | 1 | 406,267 | 178,221 | 13% | 0.03% |
| D462 | 1.40E+07 | 86.42% | IgG | 1 | 646,809 | 504,029 | 120% | 0.22% |
| | | | | 2 | 923,103 | | | |
| | | | IgK | 1 | 57,096 | 506,989 | 36% | 0.08% |
| | | | | 2 | 967,693 | | | |
| | | | IgL | 1 | 201,566 | 402,456 | 29% | 0.06% |
| | | | | 2 | 586,568 | | | |
| D791 | 3.90E+06 | 97.50% | IgG | 1 | 126,955 | 263,848 | 226% | 0.22% |
| | | | | 2 | 776,578 | | | |
| | | | | 3* | 522,308 | | | |
| | | | IgK | 1 | 527,031 | 568,736 | 146% | 0.22% |
| | | | | 2 | 934,780 | | | |
| | | | | 3* | 1,057,984 | | | |
| | | | IgL | 1 | 47,368 | 212,922 | 55% | 0.12% |
| | | | | 2 | 387,730 | | | |
| | | | | 3* | 537,611 | | | |
| D1174 | 1.80E+06 | 100.00% | IgG | 1 | 497,094 | 220,740 | 409% | 0.22% |
| | | | | 2 | 665,981 | | | |
| | | | | 3* | 678,973 | | | |
| | | | IgK | 1 | 254,890 | 415,601 | 231% | 0.22% |
| | | | | 2 | 797,239 | | | |
| | | | | 3* | 929,145 | | | |
| | | | IgL | 1 | 228,992 | 307,415 | 171% | 0.22% |
| | | | | 2 | 453,935 | | | |
| | | | | 3* | 510,711 | | | |
| D1512 | 1.00E+07 | 79.37% | IgG | 1 | 370,198 | 155,262 | 52% | 0.12% |
| | | | IgK | 1 | 408,498 | 214,651 | 21% | 0.05% |
| | | | IgL | 1 | 93,848 | 55,785 | 6% | 0.01% |
| | | | | | averages: | 293,689 | 100% | 0.13% |

* Library replicate performed using unique molecular identifiers during cDNA synthesis step.

[†] Sequencing coverage was calculated for PBMCs using QA255 D914 B cell frequency statistics from the cell sort (*Williams et al., 2015*).

[‡] Whole-body sequencing coverage was calculated assuming 10 ml of blood was sampled from a total of 4500 ml adult blood volume.

## Results

### Longitudinal sequencing of six ADCC HIV-1 antibody lineages

We previously reported the isolation of six monoclonal antibodies (mAbs), each with potent antibody-dependent cell cytotoxicity (ADCC) activity, from a sample collected 914 days post-infection (D914) from a clade A-infected Kenyan woman (*Williams et al., 2015*; *Williams et al., 2019*). All these mAbs derived from distinct B cell lineages, with four targeting two distinct gp41 Env epitopes

**Table 2.** Clonal sequence analyses for ADCC antibody lineages.

(A, B) Characteristics and statistics of heavy (A) and light (B) chain clonal families. Percent SHM was calculated as the mutation frequency at the nucleotide level compared to the predicted naïve allele, as determined by the per subject germline inference for QA255. Statistics calculated for individual timepoints within the context of the full subject repertoire were downsampled to 50–150K sequences for computational manageability. Light chain clonal family size statistics were reported but unreliable due to overclustering caveats that are explained in the Materials and methods. Per-timepoint clonal family statistics were excluded if ≤1 clonal heavy or light chain sequences were identified. #: The VH gene used in 067 and 157 lineages was determined to be an allele not cataloged in IMGT (http://www.imgt.org/): 1–69*11 +A147G+C169TI; the 157 lineage VK gene was also a new allele: 3–11*01+T5A+T9A+T36G+G84A. †: Percent SHM for 016 light chain lineage includes a 3-nt insertion in the CDRL3. SHM: somatic hypermutation; ints: intermediates. Longitudinal values for 'No. of unique clonal sequences identified' are plotted for heavy and light chains in Table 2; *Figure 2—figure supplements 1*, along with longitudinal values for 'Average %SHM (nt)' for heavy chains. Graphics displaying the most probable routes of antibody lineage maturation corresponding to the 'No. of ints resolved in lineage path' are available in *Figure 4* and *Figure 4—figure supplements 1–3*.

**Heavy chain**

| Lineage | VH | DH | JH | %SHM mature Ab | # sampled clonal seqs | Avg clonal family %SHM | D-119 | D462 | D791 | D1174 | D1512 | D-119 | D462 | D791 | D1174 | D1512 | D462 | D791 | D1174 | D1512 | No. ints resolved in lineage path |
|---|---|---|---|---|---|---|---|---|---|---|---|---|---|---|---|---|---|---|---|---|---|
| | Mature antibody statistics — Gene usage (partis) | | | | Clonal family (all timepoints merged, not downsampled) | | Clonal family statistics within repertoire (downsampled analyses) — No. unique clonal sequences identified | | | | | Size: percent of sampled repertoire | | | | | Average %SHM (nt) | | | | |
| 006 | 3–23*01 | 3–22*01 | 4*02 | 8.7 | 7 | 10.8 | 0 | 7 | 0 | 0 | 0 | | 0.002% | | | | 7.6 | | | | 1 |
| 016 | 4–34*01 | 6–13*01 | 5*02 | 9.9 | 702 | 9.2 | 0 | 581 | 112 | 0 | 9 | | 0.13% | 0.05% | | 0.01% | 9.6 | 10.3 | | 12.6 | 14 |
| 067 | 1–69*11# | 1–1*01 | 6*03 | 8.1 | 36 | 11.9 | 0 | 35 | 1 | 0 | 0 | | 0.01% | | | | 11.6 | | | | 2 |
| 072 | 1–69*11 | 4–17*01 | 3*02 | 11.5 | 515 | 12.5 | 0 | 11 | 118 | 325 | 61 | | 0.04% | 0.05% | 0.17% | 0.04% | 9.5 | 12.8 | 13.0 | 15.1 | 5 |
| 105 | 3–15*01 | 3–22*01 | 6*03 | 9.5 | 193 | 4.9 | 0 | 193 | 0 | 0 | 0 | | 0.04% | | | | 5 | | | | 2 |
| 157 | 1–69*11# | 1–26*01 | 6*03 | 6.5 | 183 | 7.4 | 0 | 6 | 165 | 8 | 4 | | 0.004% | 0.08% | 0.04% | 0.03% | 9.3 | 7.7 | 10.3 | 11.8 | 4 |

**Light chain**

| Lineage | VK | VL | JK/L | %SHM mature Ab | # sampled clonal seqs | Avg clonal family %SHM | D-119 | D462 | D791 | D1174 | D1512 | D-119 | D462 | D791 | D1174 | D1512 | D462 | D791 | D1174 | D1512 | No. ints resolved in lineage path |
|---|---|---|---|---|---|---|---|---|---|---|---|---|---|---|---|---|---|---|---|---|---|
| | Mature antibody statistics — Gene usage (partis) | | | | Clonal family (all timepoints merged, not downsampled) | | Clonal family statistics within repertoire (downsampled analyses) — No. unique clonal sequences identified | | | | | Size: percent of repertoire | | | | | Average %SHM (nt) | | | | |
| 006 | 2–11*01 | | 2*01 | 6.8 | 673 | 6.1 | 12 | 274 | 273 | 52 | 62 | | 0.14% | | | | 7.1 | | | | 7 |
| 016 | | 1–51*02 | 2*01 | 9.9† | 1518 | 7.3 | 299 | 339 | 523 | 321 | 36 | | 0.03% | 0.02% | | 0.11% | 6.6 | 7.1 | | 7.7 | 11 |
| 067 | 2–11*01 | | 3*02 | 3.3 | 20,337 | 6.5 | 2149 | 6105 | 3408 | 7946 | 729 | | 2.04% | | | | 6.6 | | | | 2 |
| 072 | 1–27*01 | | 1*01 | 7.8 | 2796 | 6.2 | 476 | 719 | 702 | 543 | 356 | | 0.17% | 0.18% | 0.18% | 0.25% | 6.3 | 6.3 | 6.6 | 7.0 | 5 |
| 105 | 3–20*01 | | 2*01 | 7.1 | 30,286 | 8.0 | 3058 | 4884 | 5607 | 14271 | 2466 | | 1.17% | | | | 6.3 | | | | 7 |
| 157 | 3–11*01 | | 5*01 | 5.0 | 18 | 8.2 | 0 | 12 | 0 | 1 | 5 | | 0.24% | | | 0.07% | 5.9 | | | 5.3 | 7 |

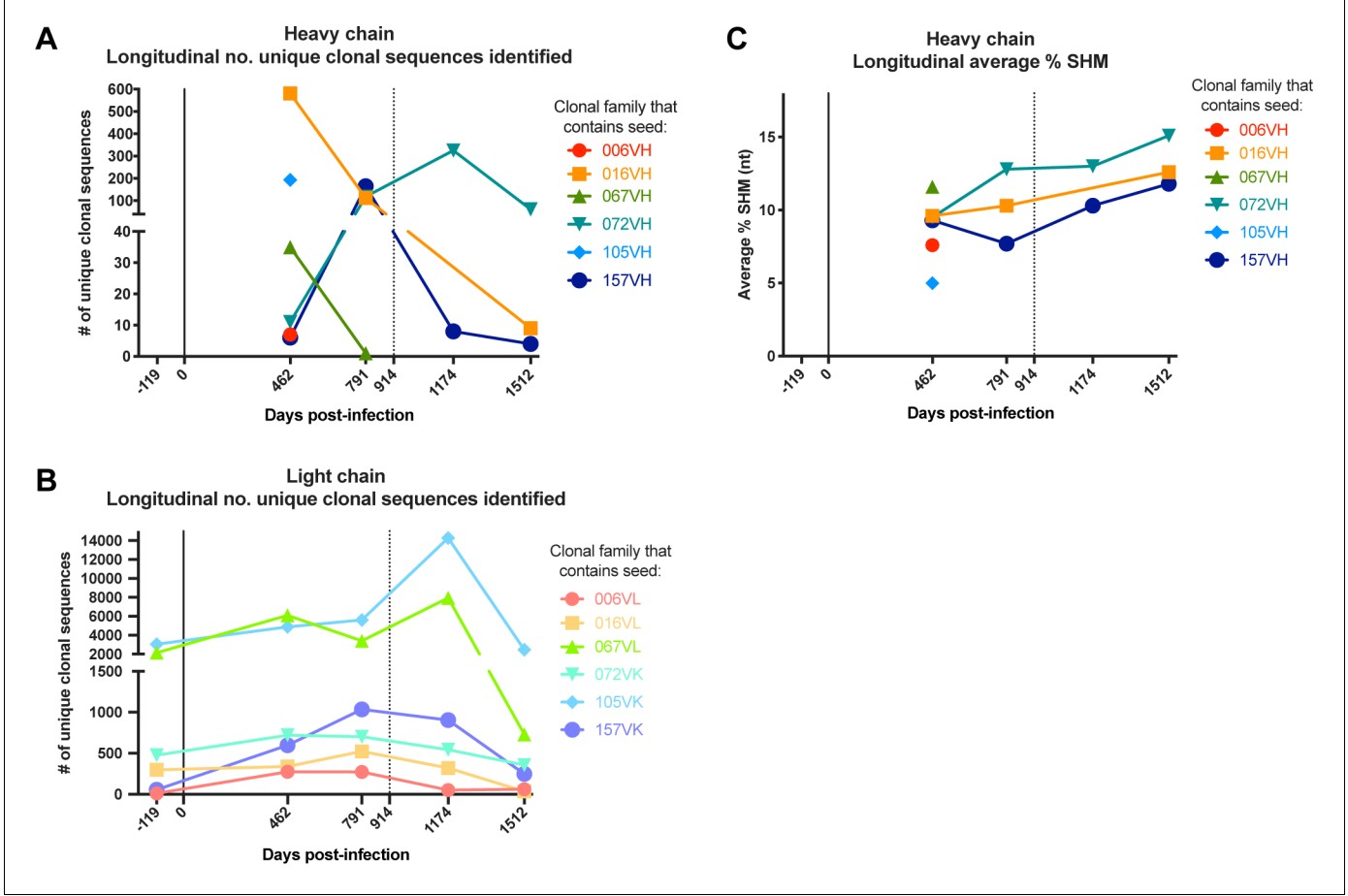

**Figure 1.** Longitudinal characteristics of heavy and light chain clonal families. (A, B) Number of clonal heavy (A) and light (B) chain sequences identified per sequenced timepoint per lineage. Caveats relating to light chain clusters are described in Methods. (C) Average variable region mutation for each clonal heavy chain family per timepoint. Dashed line at D914 indicates the timepoint of isolation of mature antibodies (*Williams et al., 2015*; *Williams et al., 2019*). SHM: somatic hypermutation.

(006, 016, 067, 072), and the other two targeting distinct gp120 Env epitopes (105, 157). One of the gp120 targeting mAbs (105) also had modest capacity to neutralize cell-free virus (*Williams et al., 2015*), whereas the others were non-neutralizing. To infer the ontogeny of these six mAbs, we used next-generation sequencing (NGS) to sequence full-length antibody variable region genes from five longitudinal blood samples collected from the subject, spanning from pre-infection (D-119) to over 4 years post-infection (D1512). Amongst the different antibody chains and timepoints sequenced, the sequencing libraries ranged widely in their sample coverage, between 6 and 409% with an average of 100% coverage, based on estimated B cell frequencies within the available peripheral blood mononuclear cell (PBMC) samples (*Table 1*). It is important to note that, despite the name 'deep sequencing', NGS-sampled datasets are relatively shallow compared to the entire repertoire. This is for two reasons: (1) each 10 mL PBMC sample is only ~0.22% of the subject's whole-body circulating blood volume (~4.5 L), not counting lymphoid organs or peripheral tissue and (2) circulating human memory B cell repertoires fluctuate over time (*Horns et al., 2019*; *Laserson et al., 2014*), making it difficult to accurately track clonal B cell families. Despite these limitations inherent to studies of this type, our sequencing efforts successfully returned clonal sequences for all six lineages of interest.

We identified sequences clonally related to the gamma heavy (VH) and lambda (VL) or kappa (VK) light chains of the six mature mAbs (*Table 2*; *Figure 1A–B*), inferred the most likely naive ancestors of each mAb chain, annotated clonal family V, (D), and J gene usage (*Table 2*), and calculated the degree of mutation in each lineage over time (*Table 2*; *Figure 1C*). Overall, the level of SHM in the mature D914 mAbs ranged from 3.3–11.5%, similar to our previous reporting of these lineages using

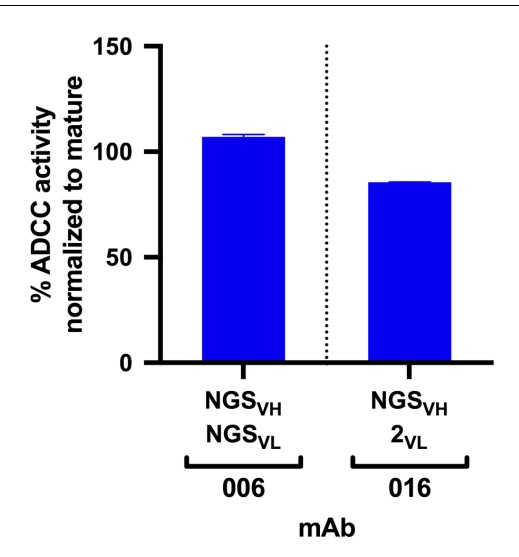

**Figure 2.** ADCC activity in antibodies made from NGS-sampled sequences. RFADCC activity of antibodies that use NGS-sampled heavy chains that are most similar in sequence to the mature 006 or 016 antibodies. The 006-NGSVH was paired with an NGS-sampled light chain; the 016-NGSVH was paired with a computationally-inferred mature-like sequence 016-2VL. mAb pairings and their mutations are illustrated in *Figure 2—figure supplement 1*, with sequences available in *Supplementary file 1*. Data reflect two independent experiments. Source data is available in *Figure 2—source data 1*.
The online version of this article includes the following source data and figure supplement(s) for figure 2:

**Source data 1.** Source data (both replicates) for RFADCC assessment of NGS mAbs, processed as detailed in Materials and methods.
**Figure supplement 1.** Graphic summaries of antibodies comprised of paired VH and VL NGS-sampled sequences.

different annotation methods (*Williams et al., 2015*; *Williams et al., 2019*). Incidentally, consensus among clonal sequences within families revealed potentially artifactual cloning mutations at the 5′ and 3′ ends of the variable regions in some of the previously characterized mature mAbs (*Williams et al., 2015*; *Williams et al., 2019*) and suggested, overall, that the consensus sequences were the most likely sequences for the antibodies. When directly compared, the sequence-corrected mAbs demonstrated equivalent epitope binding strength and ADCC activity to those previously studied (data not shown). Thus, we corrected the 5′ and 3′ ends of the mature D914 mAb variable regions to consensus sequences among each clonal family for use in this study (*Supplementary file 1*).

## mAbs constructed from D462 NGS sequences mediated ADCC

Each lineage varied in the time point(s) from which we identified clonal sequences, with most clonal VH sequences existing in the D462 datasets for four of the six lineages (*Table 2A*). The NGS sequences within each lineage that shared the highest nucleotide (nt) identity with the D914 mature VH variable regions, ranging from 88–99%, were found within the D462 or D791 timepoints (*Supplementary file 2*). As test cases, we synthesized antibodies with VH sequences derived from the D462 NGS results that were similar to the mature 006 and 016 mAbs (88.5% and 97.5% nt identity, respectively). The 006-NGS$_{VH}$ was paired with 006-NGS$_{VL}$ light chain that had 95.8% nt identity to the mature 006 VL (*Figure 2—figure supplement 1A*) and, since this option was not available for 016 VL, the 016-NGS$_{VH}$ was paired with a computationally-inferred light chain (016-2$_{VL}$) that preceded the mature in development (*Figure 2—figure supplement 1B*). mAb functionalities were then assessed using the Rapid and Fluorometric ADCC assay, which uses primary PBMCs as target and effector cells (*Gómez-Román et al., 2006*); in this method, mAbs bind to antigen-coated target cells and facilitate lysis by effector cells. Both NGS-based mAbs demonstrated full ADCC function (*Figure 2*), suggesting that ADCC function likely developed within these lineages by D462 post-infection.

### Inferred naive mAbs vary in antigen binding capability and ADCC function

Inferred naïve ancestor mAbs across lineages varied in their ability to bind HIV Env and facilitate ADCC function. Of the gp41-targeted lineages, only the inferred naive precursor of the 072 lineage bound gp41 ($K_D$ = 34.4 nM) and mediated detectable ADCC, although the activity was very low (*Figure 3A, C*). Both naive mAbs from the gp120-targeted lineages bound monomeric gp120 ($K_D$ = 23.4 nM for 105-0$_{VH}$0$_{VK}$ and 41.1 μM for 157-0$_{VH}$0$_{VK}$; *Figure 3B*). Interestingly, despite the 105 naïve mAb having higher binding affinity compared to 157 naïve mAb, it did not mediate ADCC, while the 157 naive mAb mediated potent ADCC (*Figure 3C*).

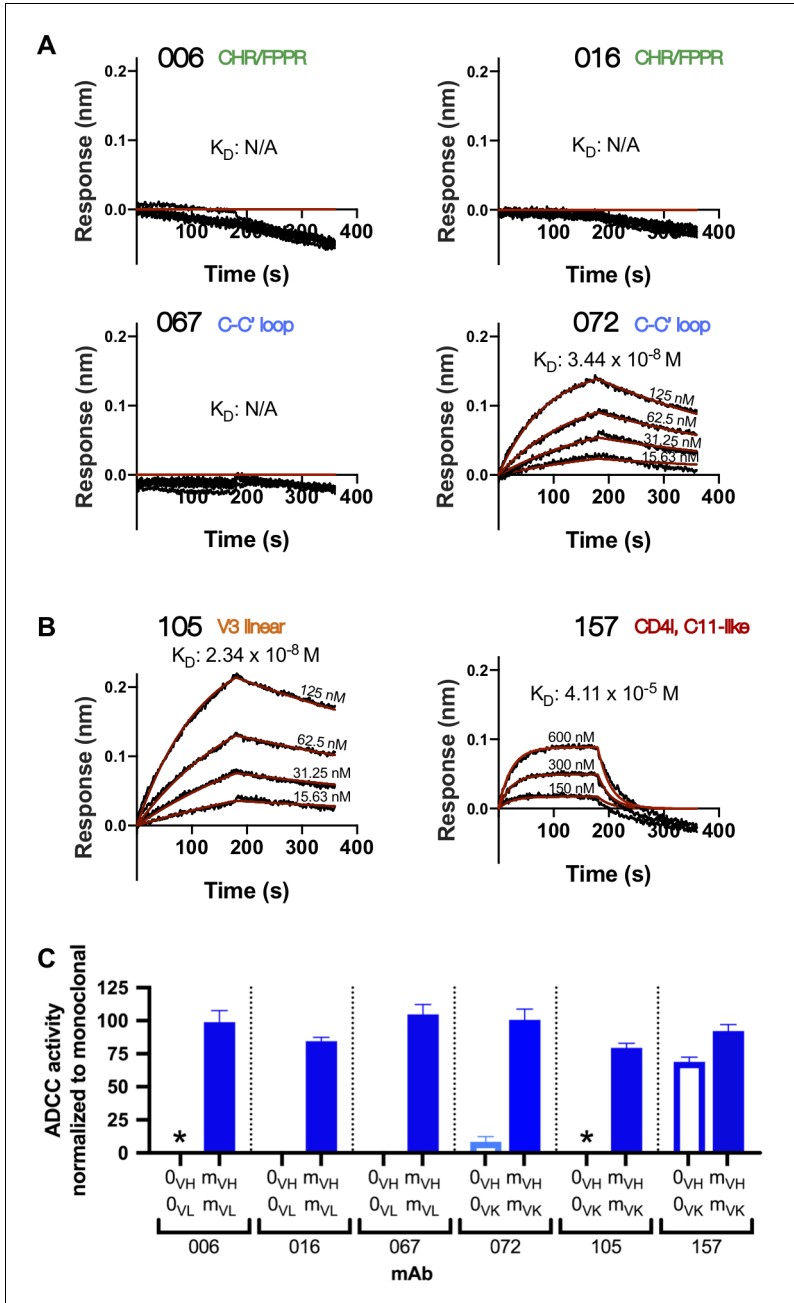

**Figure 3.** Inferred naïve mAbs from six lineages vary in antigen binding capability and ADCC function. (**A–B**) Binding kinetics of the inferred naive antibody (ligand) from each lineage to indicated concentrations of monomeric C.ZA.1197MB gp41 ectodomain (**A**) or BL035.W6M.C1 gp120 (**B**) (analyte). Best fitting lines (red) to a 1:1 binding model of ligand:analyte binding are shown. Data are representative of two independent experiments. (**C**) Positive control-normalized RFADCC activity of inferred naive antibodies compared to their respective mature antibodies. Normalization is described in Methods. Asterisks indicate indeterminate activity, as defined in Methods. Data are represented as mean ± SEM and reflect at least five independent experiments; source data are available in *Figure 3—source data 1*. 0: naive; m: mature. See *Figure 3—figure supplement 1* for functional assessment of alternative naïve mAbs.

The online version of this article includes the following source data and figure supplement(s) for figure 3:

**Source data 1.** Source data (all replicates) for RFADCC assessment of inferred naive mAbs (*Figure 3C*), processed as detailed in Materials and methods.

**Figure supplement 1.** Alternative naive mAb functionality.

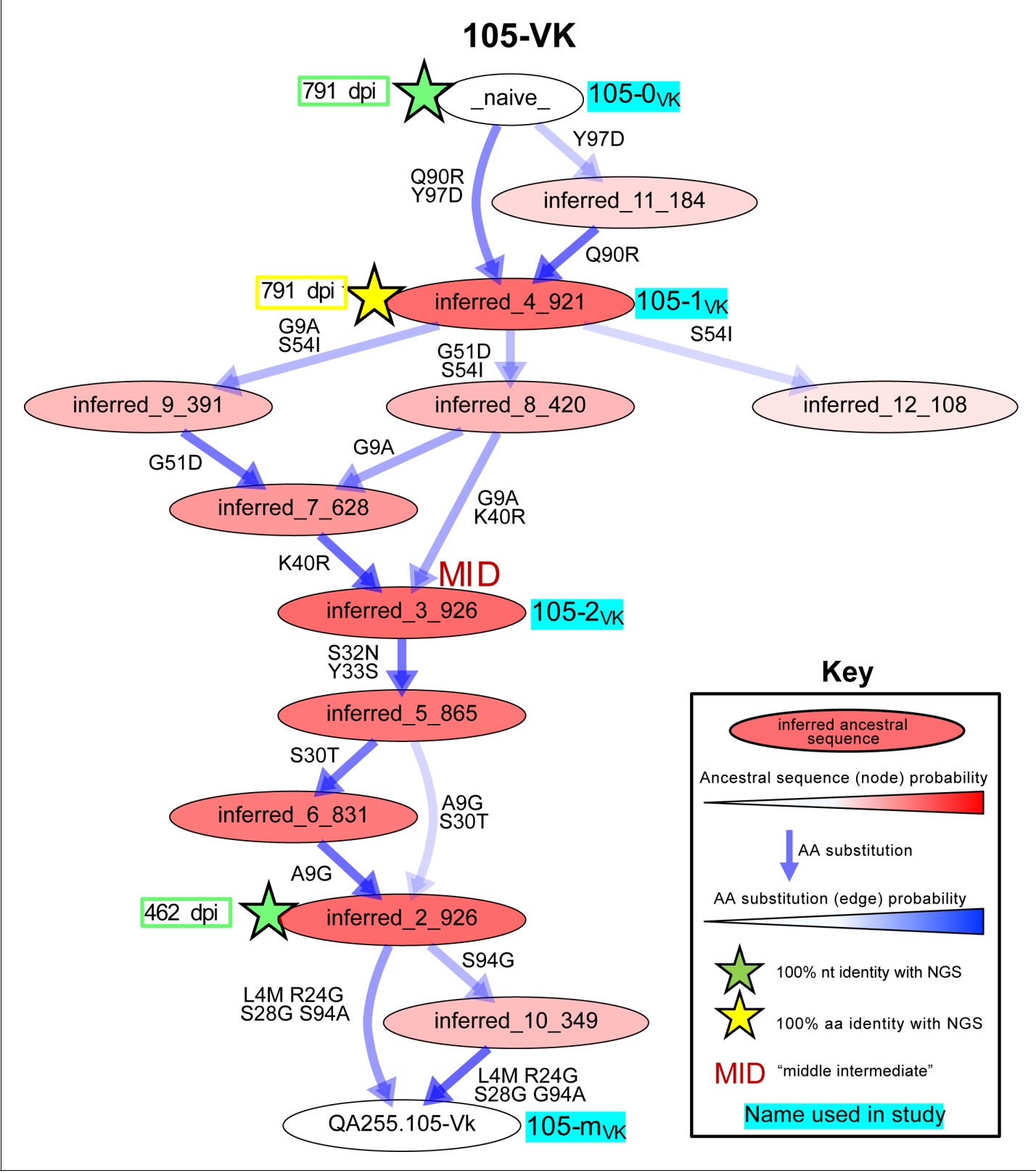

**Figure 4.** Example of most probable routes of antibody maturation: 105 light chain. Ancestral sequences (internal nodes within phylogenies) and their relative confidences were summarized for the 105 antibody light (kappa) chain from Bayesian clonal family phylogenies sampled from the associated posterior distribution for the 105-light clonal family of sampled NGS sequences. Results are displayed in this graphic illustrating multiple possible lineages of amino acid transitions and their relative confidences for 105 light chain development. Amino acid substitutions (arrows) connect the inferred

*Figure 4 continued on next page*

*Figure 4 continued*

naive sequence (white node, top) to the mature sequence (white node, bottom) via reconstructed ancestral intermediate sequences (red-shaded nodes). The red shading of nodes is proportional to the posterior probability that this ancestral sequence was present in the lineage. Low probability nodes were filtered out, resulting in some incomplete pathways within the graphics. For a given node, the blue shadings of multiple arrows arising from that node is proportional to the corresponding transition probability from that node to downstream nodes. Stars denote inferred sequences that were identical in nucleotide (green) or amino acid (yellow) sequence to sampled NGS sequences. 'MID' (red) denotes middle intermediate sequence. Labels with cyan highlighting denote sequences that were used in functionality assays (see also *Supplementary file 1*). See *Figure 4—figure supplements 1– 3* for probable maturation graphics for all 12 antibody chains of interest.

The online version of this article includes the following figure supplement(s) for figure 4:

**Figure supplement 1.** Most probable routes of antibody lineage maturation for lineages 006 and 016.
**Figure supplement 2.** Most probable routes of antibody lineage maturation for lineages 067 and 072.
**Figure supplement 3.** Most probable routes of antibody lineage maturation for lineages 105 and 157.

Typically, the uncertainty in the computational inference of antibody lineage naive precursors is largely ignored, even though single amino acid differences in the naive mAb can result in differences in HIV Env binding and functional properties (*Yuan et al., 2011*). To account for inference uncertainty, we also tested 'alternative naive' mAbs for each lineage that were inferred not by partis (*Ralph and Matsen, 2016b*), but by linearham (*Dhar et al., 2020*), a second computational method that jointly models V(D)J recombination and evolutionary history. For 7 of 12 antibody chains, this alternative approach generated naïve sequences that differed from the original naïve sequences by one to four amino acids within the CDR3 and/or FWR4 regions (*Supplementary file 1*). For the remaining five, the alternative method inferred the same naive sequence as the original method. When tested, the alternative naïve sequences did not alter antigen binding or function in five of six antibody lineages when compared to the original naïve mAbs (*Figure 3—figure supplement 1A*). However, in the 157 lineage we found that the alternative naïve VK chain (157-A$_{VK}$) differed from the original naïve (157-0$_{VK}$) by two adjacent amino acids in the CDRL3 (*Figure 3—figure supplement 1D*). When 157-A$_{VK}$ was paired with either the original (157-0$_{VH}$) or alternative (157-A$_{VH}$) VH chains, it abolished gp120 binding and ADCC functionality (*Figure 3—figure supplement 1B,C*). Thus, there is some uncertainty about whether the true 157 lineage naïve precursor was specific for HIV and/or capable of mediating ADCC.

## Characterization of lineage intermediates that confer HIV specificity and ADCC function

To reconstruct probable developmental routes that led to the mature D914 mAb heavy and light chains, we used Bayesian phylogenetic lineage inference, which emphasizes only the inferred ancestral intermediates that have high relative statistical confidence and infers chronological ordering of these likely ancestral sequences. This method was specifically developed to reconstruct antibody evolution using sparse data (*Simonich et al., 2019*). Among the 12 heavy and light chain lineages for the six mAbs, our methods resolved anywhere from 1 to 14 probable intermediate sequences that lay between the naive and the mature sequences (*Table 2*). *Figure 4*, showing 105VK development, exemplifies one pathway, although all pathways may be viewed in *Figure 4—figure supplements 1–3*. Of note, 8 of 12 of these computationally-inferred lineages were validated by the existence of NGS-sampled sequences that were identical either in nucleotide or amino acid sequence to at least one of the inferred intermediates (*Figure 4*, *Figure 4—figure supplements 1– 3*).

To focus our subsequent lineage studies on determining which heavy chain and/or light chain mutations enabled HIV Env binding and ADCC gain-of-function, we employed the following strategy to select inferred intermediates of interest (detailed fully in Materials and methods). For each chain, we chose lineage intermediates with high relative confidence that were chronologically near the middle of the inferred lineage, paired them together (see Materials and methods), and performed preliminary experiments to determine whether antigen binding or ADCC gain-of-function occurred before or after these 'middle' intermediates (*Figure 4*, *Figure 4—figure supplements 1–3*). If the middle intermediate had function, we focused on pre-middle inferred intermediates, if available within the computationally-inferred lineages. If the middle intermediate lacked function, we focused on post-middle inferred intermediates. Selected heavy and light chain sequences, were paired

together in all possible combinations, roughly chronologically (i.e. increasing SHM and within per-chain resolved chronologies), to reveal per-chain mutations contributing to mAb function. Because we did not use paired sequencing methods, these pairings do not necessarily reflect true biological intermediates, but, instead, allowed us to identify ordered steps within each chain's maturation that conferred HIV binding and/or ADCC activity. For clarity, we are reporting only the pairings that revealed steps in gain-of-function for the six lineages. Pertinent mAbs thus defined the mutations that conferred HIV binding and ADCC activity in each lineage as follows: *gp41 antibody 006*: mAb 006 (VH3-23, VL2-11) targets a discontinuous epitope that includes the C-terminal heptad repeat (*Williams et al., 2019*). Mutations in CDRH2 and CDRH3, totaling 1.4% VH SHM, were minimally required for antigen binding and ADCC activity by 006-$1_{VH}0_{VL}$ in the 006 lineage (*Figure 5A*). The addition of six light chain mutations among CDRL1, CDRL3, and FWRL2 augmented gp41 binding and ADCC potency in 006-$1_{VH}1_{VL}$.

*gp41 antibody 016*: mAb 016 (VH4-34, VL1-51) targets a similar gp41 epitope as 006 (*Williams et al., 2019*). Detectable gp41 binding and ADCC function was demonstrated by the 016-$1_{VH}1_{VL}$ mAb which contains mutations in heavy and light chain CDRs and FWRs (5.6% VH SHM and 3.6% VL SHM) (*Figure 5B*). A subsequent CDRL3 insertion of residue S94 (see details on chronology resolution of this insertion in Methods) allowed for augmented ADCC capacity (016-$1_{VH}2_{VL}$). Additional FWR and CDR mutations in the VH chain (016-$2_{VH}2_{VL}$) further strengthened gp41 binding and ADCC functionality.

*gp41 antibody 067*: mAb 067 (VH1-69, VL2-11) targets the C-C' loop (*Williams et al., 2019*). For the 067 lineage, the 067-$0_{VH}1_{VL}$ mAb incorporated a single CDRL3 mutation (0.3% VL SHM) and displayed detectable antigen binding; we could not determine whether there was meaningful ADCC function or not as levels were just above background (*Figure 5C*). To achieve more convincing ADCC function, two CDRH2 mutations were also required (1.0% VH SHM). Further mutations in either the heavy or light chain FWRs modestly increased ADCC potency to >50% that of the 067 mature mAb.

*gp41 antibody 072*: mAb 072 (VH1-69, VK1-27) targets an overlapping but distinct epitope as 067 (*Williams et al., 2019*). Unlike the other gp41-targeting lineages, the 072 lineage inferred naive mAb bound gp41 antigen and mediated weak ADCC, as aforementioned (*Figures 3A,C* and *5D*). To gain ADCC potency, only mutations in the VH chain were necessary; mAbs 072-$1_{VH}0_{VK}$ (0.8% VH SHM) and 072-$2_{VH}0_{VK}$ (0.5% VH SHM) demonstrate that two substitutions in either the CDRH1 or CDRH2 increased activity but, when combined in 072-$3_{VH}0_{VK}$, they were not additive or synergistic (*Figure 5D*). Instead, FWRH3 and FWRH4 mutations in mAb 072-$4_{VH}0_{VK}$ were required for ADCC activity comparable to that of 072 mature mAb.

*gp120 antibody 105*: mAb 105 (VH3-15, VK3-20) targets a V3 linear epitope (*Williams et al., 2015*). The inferred 105 lineage naive and 105-$0_{VH}1_{VK}$ mAbs bound gp120 but displayed only indeterminate levels of ADCC activity, as defined in Materials and methods (*Figure 6A*). 105-$1_{VH}1_{VK}$ mAb demonstrates that heavy chain CDRH1 and CDRH2 mutations (1.9% VH SHM) were required for ADCC gain of function, even though these mutations did not affect binding affinity (*Figure 6A*). mAb 105-$1_{VH}1_{VK}$ lacked full breadth, however, in that it mediated ADCC against clade A HIV strain BG505.W6M.B1, but not clade C CAP210.2.00.E8 (*Figure 6—figure supplement 1A*). ADCC breadth matching that of the 105 mature was achieved, instead, by 105-$1_{VH}2_{VK}$. Lastly, we note that the FWR4 mutation in the 105-$1_{VH}$ chain was likely a computational artifact and unlikely to affect mAb function, as this mutation is absent in the mature 105 mAb.

*gp120 antibody 157*: mAb 157 (VH1-69, VK3-11) targets a CD4-induced, C11-like epitope (*Williams et al., 2015*). As detailed in *Figure 3—figure supplement 1* and further shown in *Figure 6B*, inferred naives 157-$0_{VH}A_{VK}$ and 157-$0_{VH}0_{VK}$ differed in their binding and ADCC capabilities, leaving us uncertain about whether this lineage was able to bind gp120 and facilitate ADCC since inception or gained these capacities following two CDRL3 mutations (*Figure 6B*). The 157-$0_{VH}0_{VK}$ naive lacked full ADCC breadth, however. While this mAb demonstrated high (>50%) ADCC potency against two clade A HIV strains (*Figure 6B* and *Figure 6—figure supplement 1B*), it had low functionality against autologous QA255 transmitted founder virus and a clade C strain, and failed to mediate any ADCC against two other strains of clades C/D and B (*Figure 6—figure supplement 1B*).

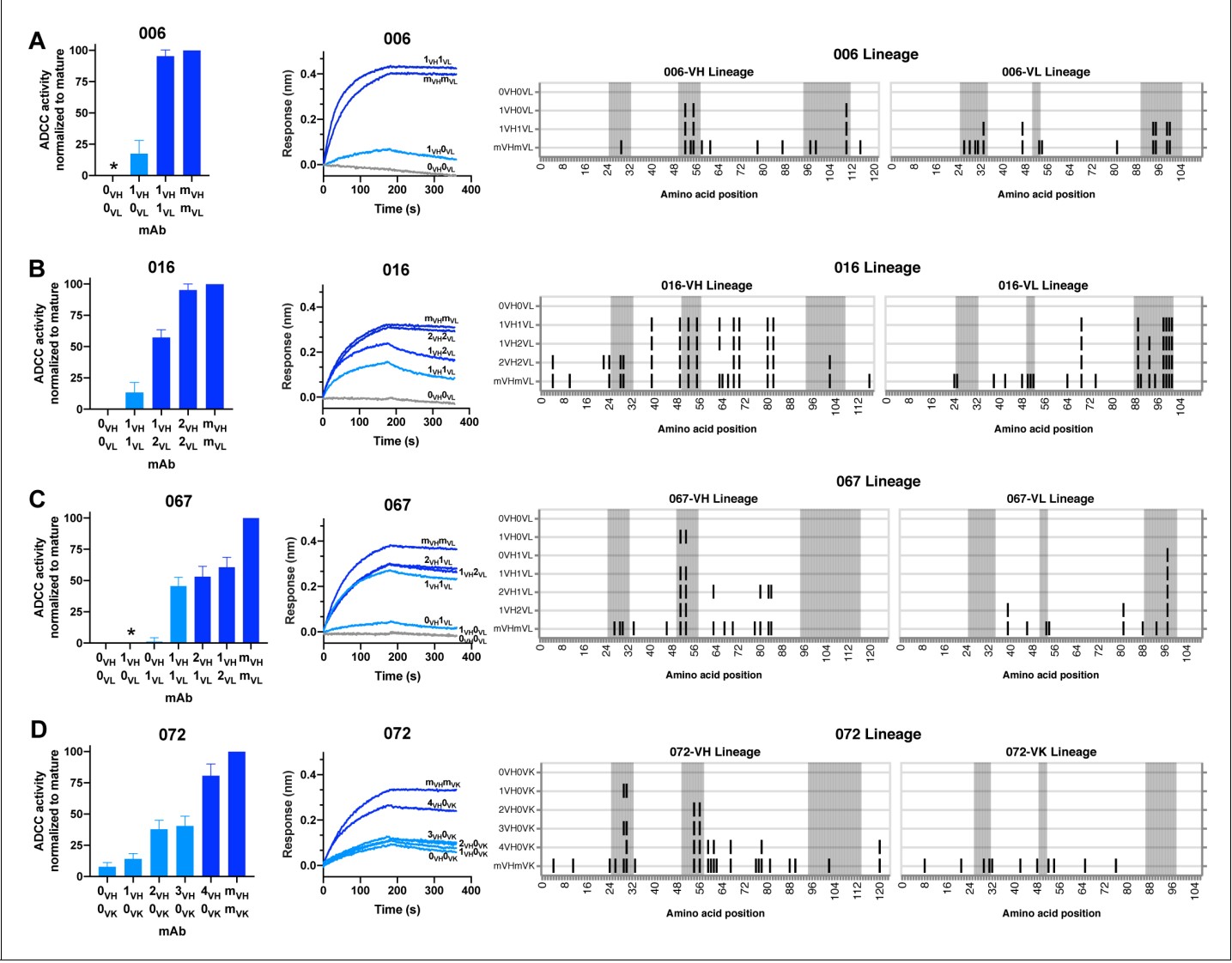

**Figure 5.** HIV specificity and ADCC development in gp41-targeted antibody lineages. (Left) Mature-normalized RFADCC activity of antibody lineage members for lineages 006 (A), 016 (B), 067 (C), and 072 (D), colored by strength of functionality: no function or indeterminate (gray), low function (<50% of mature activity; light blue), high function (>50% of mature activity; dark blue). Intermediate sequences are numbered consecutively based on their position in the developmental pathway between naïve (0) and mature (m). Asterisks indicate indeterminate activity, as defined in Methods. Data are represented as mean ± SEM and reflect at least four independent experiments, including data presented in *Figure 3* for naive and mature Abs to best account for assay variability and to compare intermediates directly to these antibodies. Source data for all replicates are available in *Figure 5—source data 1*. (Middle) Binding of lineage members (ligand) to monomeric C.ZA.1197MB gp41 ectodomain (analyte, 62.5 nM), measured by BLI. Data are representative of two independent experiments. Data are colored based on each antibody's RFADCC functionality. (Right) mAb heavy and light chain pairings illustrating variable region amino acid substitutions with respect to the reference sequence at the top of each lineage: black lines indicate amino acid substitutions; gray shaded regions demarcate CDRs.

The online version of this article includes the following source data for figure 5:

**Source data 1.** Source data (all replicates) for RFADCC assessment of Ab lineages (*Figures 5* and *6*), processed as detailed in Methods.

## Themes in ADCC development pathways among six lineages

Overall, the majority of lineages, regardless of epitope specificity, ultimately required substitutions in both the heavy and light chains to develop ADCC function >50% as potent as their respective mature mAb (*Figure 7*). In five of six lineages, detectable binding accompanied ADCC capacity, with lineage 105 being the exception that bound prior to developing ADCC function. Most notably, substitutions in CDRs were typically required for ADCC gain-of-function, while subsequent FWR

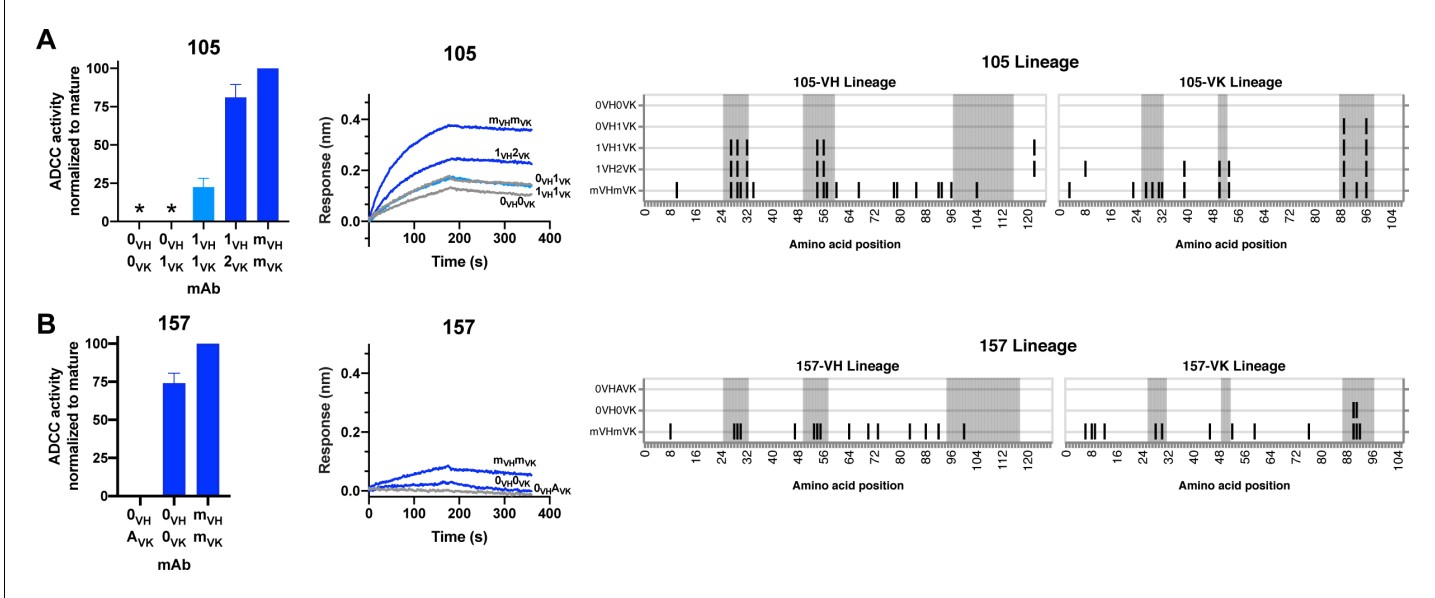

**Figure 6.** HIV specificity and ADCC development in gp120-targeted antibody lineages. RFADCC activity (left), antigen binding to monomeric BL035. W6M.C1 (analyte, 62.5 nM) as measured by BLI (middle), and mAb heavy and light chain pairings (right) for the 105 and 157 lineages all displayed as in *Figure 5*. Asterisks indicate indeterminate activity, as defined in Methods. RFADCC data are represented as mean ± SEM and reflect at least four independent experiments, including data presented in *Figure 3* for naïve and mature Abs to best account for assay variability. Source data for all replicates are available in *Figure 5—source data 1*. See also *Figure 6—figure supplement 1* and *Figure 6—source data 1* for RFADCC breadth data. The online version of this article includes the following source data and figure supplement(s) for figure 6:

**Source data 1.** Source data (all replicates) for RFADCC breadth assessment of Ab lineages 105 and 157.

**Figure supplement 1.** ADCC breadth displayed by 105 and 157 antibody lineages.

substitutions, either alone or alongside additional CDR mutations, augmented this activity (*Figure 7*). Since the chronology of inferred ancestral sequences was only resolved on a per-chain basis, we note that heavy and light chain mutations likely co-occurred in an interlaced fashion. However, in two cases (016 and 105), boosted ADCC potency was conferred by specific mutations in a single chain that were resolved to have occurred subsequent to mutations already incorporated in the prior intermediate. In other cases, namely 006, 067, and 072, where we cannot know if FWR mutations occurred before or after critical CDR mutations, we can only conclude that the FWR mutations are required in addition to CDR mutations to boost ADCC potency. A graphical summary of the key developmental steps for each lineage is presented in *Figure 7—figure supplement 1*.

## Antigen binding affinity and ADCC function correlate

In our detailed analyses of binding and ADCC activity, there were three lineages (067, 072, and 105) where we observed two mAbs from the same lineage having similar binding affinities but different ADCC capabilities. To visualize the relationship between binding affinity and ADCC potency, we plotted mAb binding affinities ($K_D$) against corresponding RFADCC potencies and found that these functionalities were positively correlated in all lineages (*Figure 8*). It is notable that the antibodies in the 157 lineage that mediate potent ADCC (naive 157-$0_{VH}0_{VK}$ and mature 157-$m_{VH}m_{VK}$) bound gp120 antigen with much lower affinity than antibodies with similar ADCC potency in the other five gp41- or gp120-targeted lineages. These data indicate that binding affinity alone does not dictate ADCC potency.

## Linear epitopes of ADCC Ab lineages remain stable over time

Lineages 067, 072, and 105 target defined linear epitopes and longitudinal sequence data from subject QA255 was available to evaluate evolution within these epitopes. The 067 and 072 lineages target overlapping C-C′ loop epitopes (*Williams et al., 2019*), and, amongst 28 longitudinal QA255

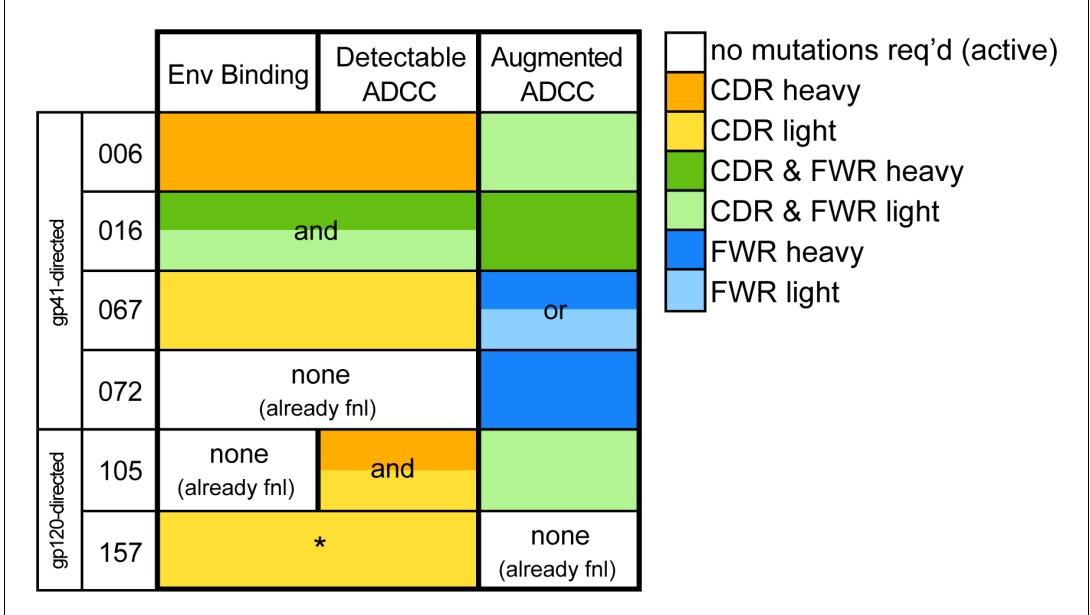

**Figure 7.** Locality of mutations required for gain of function. Summary of the location of mutations critical for acquiring binding and activity functions in all five ADCC Ab lineages, respective to the previous step, that is building upon one another to gain functions. Lineages are summarized by row. Detectable ADCC function excludes lineage members with indeterminate ADCC activity, as defined in Materials and methods. Augmented ADCC activity indicates lineage members with activity most comparable to that of their respective mature. *157-$0_{VH}A_{VK}$ required mutations for function while 157-$0_{VH}0_{VK}$ did not; Env binding and detectable ADCC determinants were defined based on this comparison. fnl: functional. See *Figure 7—figure supplement 1* for a more detailed summary graphic of key steps to attain function in all six lineages.

The online version of this article includes the following figure supplement(s) for figure 7:

**Figure supplement 1.** Key steps in ADCC development among six lineages.

Env sequences (*Bosch et al., 2010*), these epitopes remain stable between 21 dpi and 1729 dpi (Env sequences GenBank accessions MW383929-MW383956).

Lineage 105 targets the V3 loop and there were changes in this region among the QA255 Env sequences, although it was unknown if the sequence changes would impact binding of 105 to its epitope. To address this, we finely mapped the epitope and sites of escape for both the mature and naive mAbs using a phage-display deep mutational scanning (Phage-DMS) approach (*Garrett et al., 2020*). The mAbs were screened using a library displaying gp41 and V3 peptides from HIV strains BG505.W6M.C2 (clade A), BF520.W14M.C2 (clade A), and C.ZA.1197MB (clade C). Both naive and mature 105 lineage mAbs enriched (i.e. bound to) wild-type peptides generated in the backgrounds of BG505.W6M.C2 and C.ZA.1197MB Env, but not peptides of the BF520.W14M.C2 Env background (*Figure 9—figure supplement 1*). The differences in binding are likely explained by sequence differences at sites 307–309 in the V3 loop: IRI (bound) vs. VHL (not bound).

The phage display library also contained mutant peptides representing every possible single amino acid mutation in the context of all three envelope variants, allowing us to map mutations that enable escape from antibody binding. The main sites of selection indicative of escape centered around residues 308–316 spanning the sequence RIGPGQA (*Figure 9A–B*, *Figure 9—figure supplement 2*). The footprints between the naïve and mature mAbs were similar, with perhaps stronger selection observed in the naïve mAb at some positions such as 309 and 315. However, overall the data suggest that SHM and affinity maturation in this lineage did not affect the major epitope binding footprint.

We compared the longitudinal V3 sequence data (*Bosch et al., 2010*) with the detailed information on the amino acids important for binding defined by Phage-DMS. There was little evidence of strong selection pressure on the 105 epitope of Env between 21 dpi and 1729 dpi. The only sweeping mutation (H308R) emerged at 189 dpi (50% frequency) and swept the population by 560 dpi (*Figure 9C*). We infer that this substitution enabled the initial activation of the 105 lineage because

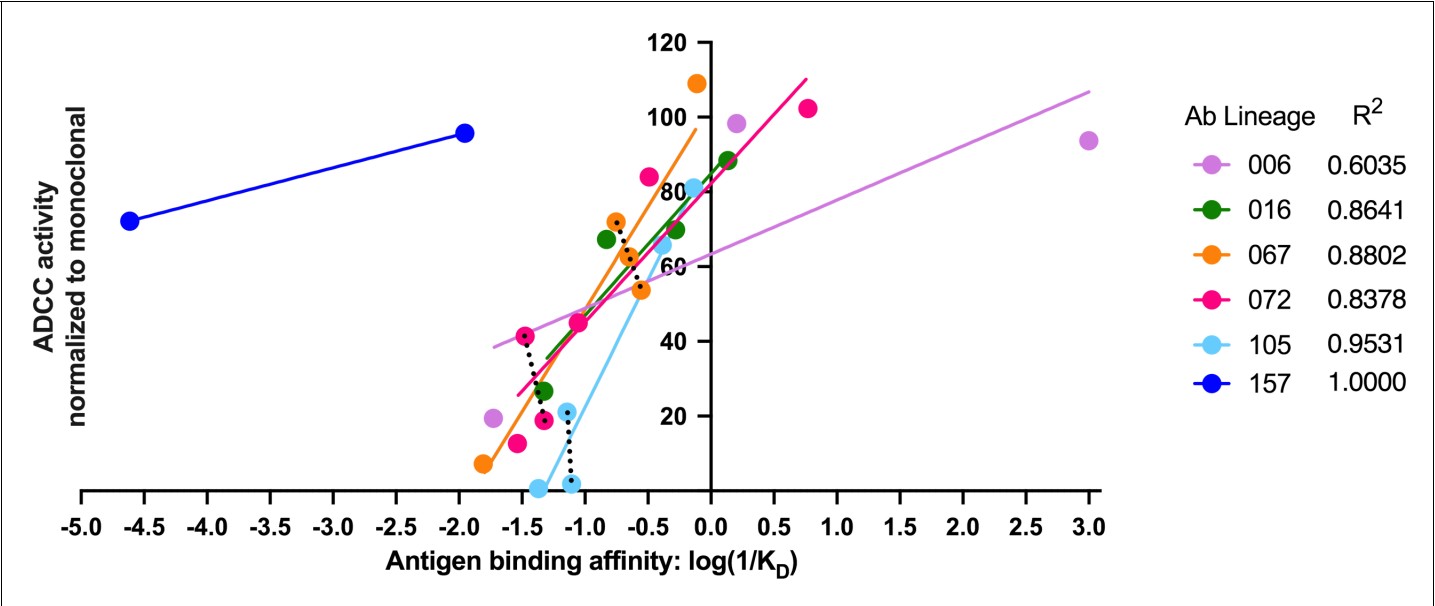

**Figure 8.** Antigen binding affinity and ADCC function correlate. For antibodies within lineages 006, 016, 067, 072, and 105 (indicated by different colors), the log of reciprocal binding affinity, as determined by representative BLI data of least two independent experiments, is plotted against positive control-normalized RFADCC activity (the data shown in *Figures 5* and *6*). Antibodies that did not detectably bind antigen by BLI are excluded as none of these mediated ADCC. Dotted lines highlight comparisons between antibodies within the same lineage that have similar binding affinities, but different RFADCC activities. Source data for each mAb are available in *Figure 8—source data 1*.

The online version of this article includes the following source data for figure 8:

**Source data 1.** Source data for correlation of binding affinity (KDs determined by BLI, see Materials and methods) and average RFADCC function (*Figures 5* and *6*; *Figure 5—source data 1*) for each mAb.

this lineage prefers R308 over H308, as aforementioned, and it is known that 105 mAbs cannot mediate ADCC against autologous transmitted founder QA255.21P.A17 gp120 (*Williams et al., 2015*), which contains H308. Except for a subset of Q315R viruses present at 189 dpi and 560 dpi, all 308–316 residues remain stable until 1729 dpi, when substitutions are observed in 7/10 clones at either residue 308 or 316, with A316T present in 50% of 1729 dpi clones (*Figure 9C*). Overall, analyses of sequence variation in the 067, 072, and 105 lineages suggest limited selection pressure for escape from ADCC antibodies in this subject.

## Discussion

Renewed interest in ADCC-capable antibodies as important to HIV vaccine responses has highlighted the need for a better understanding of their natural development (*Forthal and Finzi, 2018*). Here, we report the evolution of six ADCC antibody lineages within a single HIV-infected individual. The inferred naïve precursors of these ADCC lineages varied in their abilities to bind HIV antigen, a finding that agrees with studies of HIV-neutralizing antibody lineages (*Stamatatos et al., 2017*). To achieve potent ADCC activity, most lineages required mutations in both their heavy and light chains. There was also evidence that some of these changes contributed to greater breadth. Generally, ADCC function was achieved through mutations in CDRs, while increased ADCC potency required additional mutations in FWR. ADCC activity accompanied antigen binding in all lineages except one; in the exception, the V3/gp120-targeting 105 lineage demonstrated binding capacity prior to developing ADCC functionality. Although there was some evidence that binding affinity was not solely responsible for ADCC capability or potency, binding affinity and ADCC activity were largely correlated. In sum, this study presents six examples of developmental pathways taken by ADCC-mediating antibodies, highlighting that there are common themes in the ontogeny of HIV-specific ADCC antibodies, but that each evolutionary pathway has unique features.

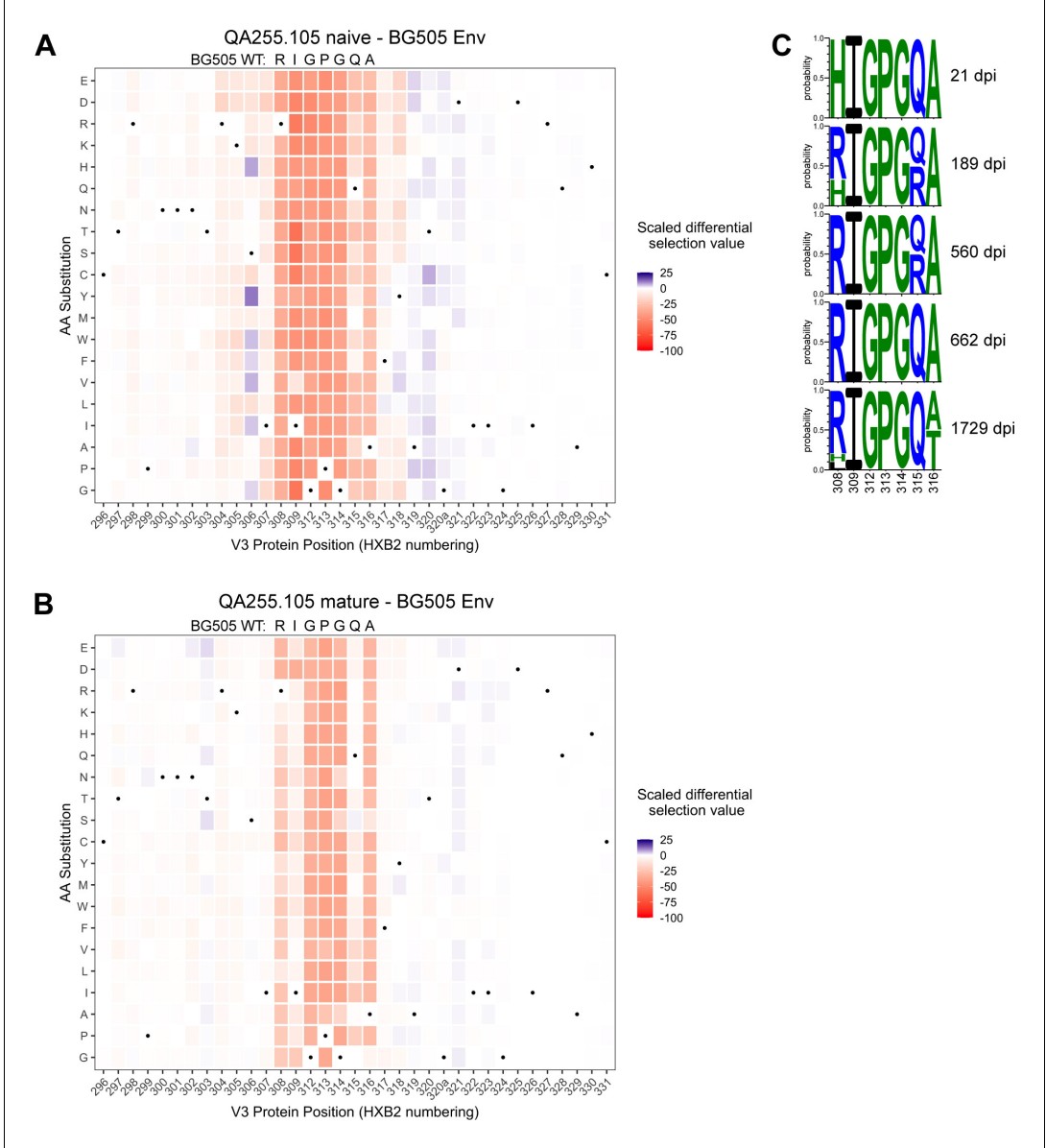

**Figure 9.** Lineage 105 epitope largely remains stable over time. (A–B) Heatmaps depict scaled differential selection: the relative effect of each possible substitution within the V3 region of Env (HXB2 numbering), compared to wild type BG505.W6M.C2 Env, on binding by (A) 105 inferred naïve mAb and (B) 105 mature mAb. Wild-type residues are indicated by black dots. Mutations enriched above the wild-type residue are colored blue while those depleted are colored red; color intensity reflects the relative amount of differential selection as indicated by the key. Data are the average of two biological replicates. See also *Figure 9—figure supplement 1* for peptide enrichment plots and *Figure 9—figure supplement 2* for scaled differential selection in C.ZA.1197MB Env. *Figure 9—source code 1* and *Figure 9—source datas 1–3* contain fold enrichment and scaled differential selection data and calculations. (C) Logo plots of the residues comprising the 105 lineage epitope amongst 28 total longitudinal QA255 Env sequences (GenBank MW383929-MW383956) using HXB2 numbering.

The online version of this article includes the following source data, source code and figure supplement(s) for figure 9:

**Source code 1.** R markdown file detailing and performing the analysis of Phage-DMS deep sequencing data in order to map antibody-targeted Env epitopes.

**Source data 1.** CSV file required for R analysis: gp41V3_PhageDMS_library_key.

**Source data 2.** CSV file required for R analysis: Phage-DMS-105epitope-Rep1annotatedCounts.

**Source data 3.** CSV file required for R analysis: Phage-DMS-105epitope-Rep2annotatedCounts.

**Figure supplement 1.** Phage-DMS peptide enrichment for 105 inferred naive and 105 mature mAbs.

**Figure supplement 2.** Lineage 105 epitope and predicted escape routes within C.ZA.1197MB Env.

Although not widely applied, antibody lineage studies require robust inference methods that account for statistical uncertainty if we are to be confident in their conclusions. This is highlighted in this study by the different properties displayed by two very similar inferred naïve precursors of the 157 lineage. The development of the 157 lineage was ambiguous: computational uncertainty in two CDRL3 residues resulted in multiple probable naive antibodies with different properties. In these different scenarios, the 157 naïve Ab was either fully capable of both antigen binding and potent ADCC, or it required two CDRL3 substitutions to achieve both functions. In the other five lineages, our use of multiple, tailored computational approaches allowed us to report results with high confidence, since the uncertainties present in computational inference of those lineages did not affect functional characteristics.

Three gp41-specific antibody lineages were not capable of binding HIV Env by their inferred naïve precursors, as measured by BLI; they developed this ability through mutation. The lack of detectable antigen binding by inferred naïve precursors is not altogether surprising as this has been observed for a number of HIV bnAbs (*Stamatatos et al., 2017*). Binding capacity was gained in two of the gp41-directed lineages (006 and 067) through limited mutations localized in the CDRs of single antibody chains (either heavy or light, respectively). The other gp41-directed lineage, 016, which targets a similar epitope as the 006 lineage, required many mutations amongst the CDRs and FWRs of both chains, highlighting that the gain of activity by different lineages can vary even if they are targeting the same epitope. As is true for all studies of this type, it is possible that our methods were not sensitive enough to detect low level binding or that, despite our best efforts, the inferred naïve sequences were inaccurate. Gp41-directed antibodies have been found to recognize self-antigens (*Verkoczy and Diaz, 2014*; *Verkoczy et al., 2014*) and this, or its response to another antigen, could explain how the 016 lineage emerged before gaining HIV-specificity.

We chose the RFADCC assay because RFADCC signal has been repeatedly correlated with disease outcome in humans (*Dhande et al., 2018*; *Lewis et al., 2019*; *Mabuka et al., 2012*; *Madhavi et al., 2014*; *Milligan et al., 2015*; *Ruiz et al., 2017*), thus making it relevant to understanding the development of potentially protective responses. Additionally, we previously demonstrated that the BL035.W6M.C1 strain of gp120 used here yields RFADCC results representative of activity against seven different strains of various clades (*Milligan et al., 2015*). There was evidence that the 072 lineage had ADCC function from inception, although the amount of ADCC was low. The remaining lineages required an average of 1.2% SHM to gain detectable ADCC function (ranging 0.1–4.3% SHM). For augmentation of ADCC function to >50% of mature activity, the lineages required an average of 2.2% SHM (range 0.6–4.8% SHM). NGS-sampled sequences that were present at 462 dpi displayed high function with 5.3% SHM (006 lineage), and developmental inferences for the 006-VH, 067-VH and 105-VH chains were informed by clonal sequences from only the 462 dpi time point, suggesting that the fully functional VH intermediates inferred for these lineages were likely present prior to 462 dpi. This is not surprising, as ADCC responses to HIV can develop within the first few months of infection (*Dugast et al., 2014*). Low mutation levels to achieve potent ADCC are promising for ADCC vaccine efforts, especially since these six characterized lineages have desirable characteristics for combating HIV, such as potent ADCC activity, cross-clade ADCC breadth, highly conserved epitopes, and potential to help clear infected cells (*Williams et al., 2015*; *Williams et al., 2019*). Moreover, these levels of mutation are easily attainable by vaccines (*Schramm and Douek, 2018*).

Importantly, most lineages required FWR mutations in one or both chains to achieve potent ADCC functionality, with or without additional CDR mutations. Although they can cost thermostability (*Henderson et al., 2019*), FWR mutations are known to contribute to the properties of HIV-specific bnAbs, predominantly by altering Ab conformation in ways that allow conserved epitopes to be bound with high affinity (*Henderson et al., 2019*; *Kepler and Wiehe, 2017*; *Ovchinnikov et al., 2018*). FWR residues contribute to the structure of the Ab and, though they do not usually directly contact the antigen, they can affect binding affinity indirectly (*Foote and Winter, 1992*). Furthermore, since they indeed affect Ab structure, FWR mutations could confer favorable Ab flexibility or an orientation that mediates better Fc receptor dimerization and ADCC potency (*Acharya et al., 2014*). FWR mutations in Fab 'elbow' regions can affect conformational flexibility and paratope plasticity during bnAb development (*Henderson et al., 2019*). Here, the 006-1_{VH} lineage intermediate caused ADCC gain-of-function and it contained, among other mutations, a CDRH3 elbow mutation

Y111F. Future structural studies are warranted to explore how FWR mutations affect Ab structure and function in these lineages.

Binding affinity has been implicated in the regulation of ADCC in several studies on tumor-specific Abs (*Mazor et al., 2016*; *Tang et al., 2007*). Here, while binding affinity correlated with ADCC function, it was not solely responsible for ADCC potency. In one lineage, binding capacity preceded detectable ADCC function. Notably, we found that half of the lineages contained clonally-related antibodies with similar binding affinity to one another but markedly different ADCC potencies. Conversely, mAbs within the two gp120-targeted lineages exemplify that there can be dramatic differences in binding affinity among mAbs that mediate equivalent, potent ADCC activity. Thus, ADCC function cannot be predicted based on affinity alone. These data support the notion that factors such as epitope specificity, binding mode, and other antibody-antigen interaction dynamics contribute to ADCC potency, as other studies have emphasized (*Acharya et al., 2014*; *Orlandi et al., 2020*; *Tolbert et al., 2020*).

Amongst Env sequences collected over ~4.5 years of HIV infection in subject QA255, we observed little directional escape in the C-C' loop and V3 epitopes targeted by the 067, 072, and 105 antibody lineages, and only at later stages of infection. This finding could indicate that selective pressure exerted by ADCC antibodies is often not enough to drive escape, although there is evidence that escape can occur (*Isitman et al., 2012*). There are at least two possible explanations for limited escape from ADCC antibodies in an infected individual: one is that the activity of ADCC antibodies is insufficient to drive escape because they have limited impact on infection dynamics; the other is that the targeted killing of infected cells leads to less pressure for escape compared to the process of blocking virus entry mediated by neutralizing antibodies. If the latter is true, ADCC could be particularly useful components of vaccines responses and therapeutic tools.

Our study offers a granular understanding of how ADCC functionality developed in six human Ab lineages. The results of this study are relevant to rational vaccine design, especially as technologies for guiding Ab development continue to emerge through reverse vaccinology efforts (*Burton, 2017*; *Rappuoli et al., 2016*; *Schramm and Douek, 2018*). Additionally, the reliance on FWR mutations to achieve high ADCC functionality could inform mAb optimization strategies for therapeutic uses.

## Materials and methods

**Key resources table**

| Reagent type (species) or resource | Designation | Source or reference | Identifiers | Additional information |
|---|---|---|---|---|
| Cell line (human, female) | FreeStyle 293 F cells | Invitrogen | Cat#R790-07; RRID:CVCL_D603 | mAb production |
| Cell line (human, female) | CEM.NKR cells; target cells | NIH AIDS Reagent Program | Cat#458; RRID:CVCL_X622 | RFADCC |
| Biological sample (human, female) | QA255 PBMCs; D-119; D462; D791; D1174; D1512 | PMID:12060878 | Subject ID:QA255 | Prospective cohort of HIV-1 negative high-risk women in Mombasa, Kenya; longitudinal PMBC samples from 119 days pre-infection, 462 days post-infection(dpi), 791 dpi, 1174 dpi, and 1512 dpi |

*Continued on next page*

Continued

| Reagent type (species) or resource | Designation | Source or reference | Identifiers | Additional information |
|---|---|---|---|---|
| Antibody (human, female) | 006-0VH0VL; 006-1VH0VL; 006-1VH1VL; 016-0VH0VL; 016-1VH1VL; 016-1VH2VL; 016-2VH2VL; 067-0VH0VL; 067-1VH0VL; 067-0VH1VL; 067-1VH1VL; 067-2VH1VL; 067-1VH2VL; 072-0VH0VK; 072-1VH0VK; 072-2VH0VK; 072-3VH0VK; 072-4VH0VK; 105-0VH0VK; 105-0VH1VK; 105-1VH1VK; 105-1VH2VK; 157-0VHAVK; 157-0VH0VK (human monoclonal) | This paper | This paper: *Supplementary file 1* | Computationally-inferred human (subject QA255) monoclonal antibodies representing naive and intermediate lineage members of six ADCC-capable antibody lineages; RFADCC (1:100-1:500); BLI (8 ug mL-1) |
| Antibody (human, female) | 006-NGS$_{VH}$NGS$_{VL}$; 016-NGS$_{VH}$2$_{VL}$ (human monoclonal) | This paper | This paper: *Supplementary file 1*; BioProject: PRJNA639297 | NGS-sampled antibody chains paired into monoclonal human (subject QA255) antibodies; 016-2$_{VL}$ light chain was computationally-inferred and not directly NGS-sampled; RFADCC (1:100-1:500) |
| Antibody (human, female) | 006-mVHmVL; 016-mVHmVL; 067-mVHmVL; 072-mVHmVK; 105-mVHmVK; 157-mVHmVK (human monoclonal) | This paper and PMIDs 30779811 and 26629541 | GenBank: MT791224-MT791235 | Functionally-isolated human (subject QA255) antibody chains; 'mature' antibodies are capable of ADCC function; some sequences were edited based on consensus NGS data as describe in this paper; RFADCC (1:100-1:500); BLI (8 ug mL-1) |
| Recombinant DNA reagent | 006-0VH; 006-0VL; 006-1VH; 006-1VL; 006-mVH; 006-mVL; 006-NGSVH; 006-NGSVL; 016-0VH; 016-0VL; 016-1VH; 016-1VL; 016-2VL; 016-2VH; 016-mVH; 016-mVL; 016-NGSVH; 067-0VH; 067-0VL; 067-1VH; 067-1VL; 067-2VH; 067-2VL; 067-mVH; 067-mVL; 072-0VH; 072-0VK; 072-1VH; 072-2VH; 072-3VH; 072-4VH; 072-mVH; 072-mVK; 105-0VH; 105-0VK; 105-1VK; 105-1VH; 105-2VK; 105-mVH; 105-mVK; 157-0VH; 157-AVK; 157-0VH; 157-0VK; 157-mVH; 157-mVK | This paper; GeneWiz | This paper: *Supplementary file 1* | Computationally-inferred or NGS-sampled human (subject QA255) IgG (VH), IgK (VK), and IgL (VL) antibody variable region sequences representing naïve, intermediate, and mature lineage members of six ADCC-capable antibody lineages |
| Recombinant DNA reagent | Human Igγ1 expression vector | PMID:17996249 | Addgene:80795 | |

*Continued*

| Reagent type (species) or resource | Designation | Source or reference | Identifiers | Additional information |
|---|---|---|---|---|
| Recombinant DNA reagent | Human Igκ expression vector | PMID:17996249 | Addgene:80796 | |
| Recombinant DNA reagent | Human Igλ expression vector | PMID:17996249 | Addgene:99575 | |
| Sequence-based reagent | NGS; NGS-sampled sequences; libraries; D-119; D462; D791; D1174; D1512 | This paper | BioProject: PRJNA639297 | Longitudinal QA255 full-length antibody variable region IgG, IgM, IgK, and IgL sequences |
| Sequence-based reagent | Primers | PMID:27525066; PMID:31097697 | see PMID:31097697 | Primers for amplification and sequencing of human IgM, IgG, IgK and IgL variable regions |
| Sequence-based reagent | SmartNNNa; template switch adaptor primers | PMID:27490633 | N/A | AAGCAGUGGTAUCA ACGCAGAGUNNNNU NNNNUNNNNUCTT rGrGrGrGrG |
| Peptide, recombinant protein | gp120 (HIV strain BL035.W6M.C1 'BL035') | Immune Technology Corp. | Cat#IT-001–115 p; GenBank: DQ208480.1 | BLI analyte (250 nM or 62.5 nM); RFADCC coat (1.5 µg per 105 cells) |
| Peptide, recombinant protein | gp120 (HIV strain QA255.22P.A17 'QA255') | Cambridge Biologics | Cat#01-01-1743; GenBank: MW383930 | RFADCC coat (1.5 µg per 105 cells) |
| Peptide, recombinant protein | gp120 (HIV strain BG505.W6M.B1 'BG505') | Cambridge Biologics | Cat#01-01-1028; GenBank: ABA61515.1 | RFADCC coat (1.5 µg per 105 cells) |
| Peptide, recombinant protein | gp120 (HIV strain CAP210.2.00.E8 'CAP210') | Immune Technology Corp. | Cat#IT-001-RC12p; GenBank: DQ435683.1 | RFADCC coat (1.5 µg per 105 cells) |
| Peptide, recombinant protein | gp120 (HIV strain MK184.W0M.G3 'MK184') | Immune Technology Corp. | Cat#IT-001–112 p; GenBank:DQ208487 | RFADCC coat (1.5 µg per 105 cells) |
| Peptide, recombinant protein | gp120 (HIV strain SF162) | Cambridge Biologics | Cat#01-01-1063; GenBank: P19550.1 | RFADCC coat (1.5 µg per 105 cells) |
| Peptide, recombinant protein | gp41 ectodomain (HIV strain C.ZA.1197MB) | Immune Technology Corp. | Cat#IT-001–0052 p; GenBank: AY463234.1 | BLI analyte (250 nM or 62.5 nM); R FADCC coat (1.5 µg per 105 cells) |
| Peptide, recombinant protein | Phage-DMS gp41 and V3 library | PMID:33089110 | N/A | Deep mutational scanning phage display library containing wildtype and mutant peptides that tile across the gp41 and V3 portions of three Envelope strains (BG505.W6M.C2, BF520.W14M.C2, and C.ZA.1197MB). Peptides are 31 amino acids long and contain every possible single amino acid mutation at the central position, with peptides overlapping by 30 amino acids. |

*Continued on next page*

*Continued*

| Reagent type (species) or resource | Designation | Source or reference | Identifiers | Additional information |
|---|---|---|---|---|
| Chemical compound, drug | Q5 High-Fidelity Master Mix | New England BioLabs | Cat#M0492S | |
| Chemical compound, drug | FreeStyle Max | Thermo Fisher Scientific | Cat#16447500 | |
| Chemical compound, drug | Protein G agarose | Pierce | Cat#20397 | |
| Chemical compound, drug | 293F FreeStyle Expression media | Invitrogen | Cat#12338–026 | |
| Commercial assay or kit | AllPrep DNA/ RNA Mini Kit | Qiagen | Cat#80204 | |
| Commercial assay or kit | SMARTer RACE 5'/3' Kit | Takara Bio USA | Cat#634858 | |
| Commercial assay or kit | KAPA library quantification kit | Kapa Biosystems | Cat#KK4824 | |
| Commercial assay or kit | 600-cycle MiSeq Reagent Kit v3 | Illumina | Cat#MS-102–3003 | |
| Commercial assay or kit | Nextera XT 96-well index kit | Illumina | Cat#FC-131–1001 | |
| Commercial assay or kit | Anti-human IgG Fc capture biosensors | Pall ForteBio | Cat#18–5063 | |
| Software, algorithm | FLASH v1.2.11 | PMID:21903629 | | http://ccb.jhu.edu/software/FLASH/ |
| Software, algorithm | Cutadapt 1.14 with Python 2.7.9 | PMID:23671333 | RRID:SCR_011841 | http://cutadapt.readthedocs.io/en/stable/ |
| Software, algorithm | FASTX toolkit 0.0.14 | Hannon Lab, Cold Spring Harbor | RRID:SCR_005534 | http://hannonlab.cshl.edu/fastx_toolkit/ |
| Software, algorithm | Partis | PMID:26751373 | N/A | https://github.com/psathyrella/partis |
| Software, algorithm | Linearham | PMID:32804924 | N/A | https://github.com/matsengrp/linearham |
| Software, algorithm | FastTree 2 | PMID:20224823 | N/A | http://doi.org/10.1371/journal.pone.0009490.g003 |
| Software, algorithm | Prune.py | PMID:31097697 | N/A | https://github.com/matsengrp/cft/blob/master/bin/prune.py |
| Software, algorithm | CFT (Clonal Family Tree) | This paper | N/A | https://github.com/matsengrp/cft/ |
| Software, algorithm | RevBayes | PMID:27235697 | N/A | https://revbayes.github.io/; incorporated in Ecgtheow code |
| Software, algorithm | Ecgtheow | PMID:31097697 | N/A | https://github.com/matsengrp/ecgtheow |
| Software, algorithm | Local BLAST for lineage-like NGS sequences | This paper | N/A | https://git.io/Je7Zp |

*Continued on next page*

*Continued*

| Reagent type (species) or resource | Designation | Source or reference | Identifiers | Additional information |
|---|---|---|---|---|
| Software, algorithm | FlowJo v10 | TreeStar | RRID:SCR_008520 | |
| Software, algorithm | Excel | Microsoft Office | RRID:SCR_016137 | |
| Software, algorithm | ForteBio's Octet Software 'Data Analysis 7.0' | Pall ForteBio | N/A | |
| Software, algorithm | Prism 8.0 c | GraphPad | RRID:SCR_002798 | |
| Software, algorithm | Rstudio | Rstudio | RRID:SCR_000432 | |

## Human subject

Peripheral blood mononuclear cell (PBMC) samples were obtained between 1997 and 2002 from a female HIV-1 seroconverted subject, QA255, who was enrolled in a prospective cohort of HIV-1-negative high-risk women in Mombasa, Kenya (*Lavreys et al., 2002*). QA255 was 40 years old at the time of HIV infection (D0). Study participants were treated according to Kenyan National Guidelines; QA255 did not receive antiretrovirals at any point during the period in which samples were analyzed for this study. Antiretroviral therapy was offered to all participants in the Mombasa Cohort beginning in March 2004, with support from the President's Emergency Plan for AIDS Relief. The infecting virus was clade A based on envelope sequence (*Bosch et al., 2010*). Approval to conduct this study was provided by the ethical review committees of the University of Nairobi Institutional Review Board, the Fred Hutchinson Cancer Research Center Institutional Review Board, and the University of Washington Institutional Review Board. Study participants provided written informed consent prior to enrollment.

## Sample handling and RNA isolation

PBMCs stored in liquid nitrogen were thawed at 37°C, diluted 10-fold in pre-warmed RPMI and centrifuged for 10 min at 300 x *g*. Cells were washed once in phosphate-buffered saline, counted with trypan blue, centrifuged again, and total RNA was extracted from PBMCs using the AllPrep DNA/RNA Mini Kit (Qiagen, Germantown, MD), according to the manufacturer's recommended protocol. RNA was stored at −80°C.

## Sequencing of full-length antibody gene variable regions

Antibody sequencing was performed as previously described (*Simonich et al., 2019*; *Vigdorovich et al., 2016*). Library preparation was performed in technical replicate, as indicated in *Table 1*, using the same RNA isolated from each timepoint: D-119 (119 days prior to HIV infection), D462, D791, D1174, and D1512 post-infection. Briefly, RACE-ready cDNA synthesis was performed using the SMARTer RACE 5′/3′ Kit (Takara Bio USA, Inc, Mountain View, CA) using primers with specificity to IgM, IgG, IgK, and IgL, as previously reported (*Simonich et al., 2019*). One replicate each for D791 and D1174 were prepared using template switch adaptor primers that included unique molecular identifiers in the cDNA synthesis step: SmartNNNa 5′ AAGCAGUGGGTAUCAACG-CAGAGUNNNNUNNNNUNNNNUCTTrGrGrGrGrG 3′(*Turchaninova et al., 2016*), where 'rG' indicates (ribonucleoside) guanosine bases. cDNA was diluted in Tricine-EDTA according to the manufacturer's recommended protocol. First-round Ig-encoding sequence amplification (20 cycles) was performed using Q5 High-Fidelity Master Mix (New England BioLabs, Ipswich, MA) and nested gene-specific primers. Amplicons were directly used as templates for MiSeq adaption by second-round PCR amplification (10–20 cycles), purified and analyzed by gel electrophoresis, and indexed using Nextera XT P5 and P7 index sequences (Illumina, San Diego, CA) for Illumina sequencing according to the manufacturer's instructions (10 cycles). Gel-purified, indexed libraries were quantitated using the KAPA library quantification kit (Kapa Biosystems, Wilmington, MA) performed on an

Applied Biosystems 7500 Fast real-time PCR machine. Libraries were denatured and loaded onto Illumina MiSeq 600-cycle V3 cartridges, according to the manufacturer's suggested workflow.

## Sequence analysis and naïve inference

Sequences were preprocessed using FLASH, cutadapt, and FASTX-toolkit as previously described (*Simonich et al., 2019*; *Vigdorovich et al., 2016*). The sequences from our NGS replicates were merged either cumulatively or on a per-timepoint basis, as appropriate, to achieve the highest depth possible for each analysis. Sequences were then deduplicated and annotated with partis (https://github.com/psathyrella/partis) using default options including per-sample germline inference (*Ralph and Matsen, 2016a*; *Ralph and Matsen, 2016b*; *Ralph and Matsen, 2019*; ). Sequences with internal stop codons or CDR3 regions that were out-of-frame or had mutated codons at the start or end were removed. Indel events were identified, tracked, and reversed for alignment purposes. We did not exclude singletons in an attempt to retain very rare or undersampled sequences. Sequencing run statistics are detailed in *Table 1*. Cumulative and per-timepoint datasets underwent clonal family clustering using both the partis unseeded and seeded clustering methods (*Ralph and Matsen, 2016b*).

Inference of unmutated common ancestor sequences and simultaneous clonal family clustering was performed using the seed clustering method along with previously-identified QA255 mature ADCC antibody sequences (*Williams et al., 2015*; *Williams et al., 2019*) as 'seeds'. As an additional measure to ensure highest accuracy for inferred naïve sequences, the uncertainty on each inferred naive sequence was visualized both with the partis –view-alternative-annotations option and by comparing these results to the most likely naive sequences inferred by linearham software (see next section). The comparison revealed only the minor differences that would be expected based on linearham's method, which uses a more detailed model to infer more accurate naive sequences for individual clonal families. For the unseeded clustering, each dataset was subsampled to 50–150K sequences for computational efficiency. For each dataset, three random subsamples were analyzed and compared to ensure that our subsampling was sufficient to minimize statistical uncertainties. Seeded analyses were not subsampled.

Due to the lack of D genes and much shorter non-templated regions in light-chain rearrangements, computational clustering analyses artifactually overestimate clonality in light chain families, that is they cluster together sequences that did not originate from the same rearrangement event, but which come from almost identical naïve rearrangements. This caveat, affecting all unpaired antibody chain sequencing studies, ultimately reduces accuracy in inferring true light chain clonal families and their maturation pathways.

## Alternative naïve inference

Alternative naïve sequences were inferred by applying a Bayesian phylogenetic Hidden Markov Model approach using the linearham software (https://github.com/matsengrp/linearham), using the partis-inferred clonal family clusters for each lineage as input (*Dhar et al., 2020*). Linearham samples naive sequences from their posterior distribution rather than providing a single naive sequence estimate, like other software programs do. The most probable linearham-predicted naïve sequences were compared to the most probable partis-predicted naïve sequences. In families with largely symmetric phylogenetic trees, the two methods return similar results, whereas with highly imbalanced trees, linearham is much more accurate.

## Antibody lineage reconstruction

Antibody lineages were inferred as previously described (*Simonich et al., 2019*). In order to subsample large clonal families according to phylogenetic relatedness to the antibody chain of interest, initial phylogenetic trees of each QA255 clonal family was inferred using FastTree 2 (*Price et al., 2010*). This allowed clonal families to be reduced to the 75–100 sequences most relevant to inferring the lineage history of the antibody chain of interest (https://github.com/matsengrp/cft/blob/master/bin/prune.py), which was then analyzed with RevBayes (*Höhna et al., 2016*) using an unrooted tree model with the general time-reversible (GTR) substitution model. All settings for RevBayes runs were customized to ensure likelihood and estimated sample sizes were >100 for each lineage. MCMC iterations ranged from 10,000 to 200,000, with thinning frequencies between 10 and 200 iterations and

number of burn-in samples between 10 and 190. All RevBayes runs were done in technical duplicate (i.e. specifying different starting trees) and duplicates were all confirmed to agree on lineage chronology. RevBayes output was summarized for internal node sequences (https://github.com/mat-sengrp/ecgtheow), resulting in summary graphics where relative confidence in unique inferred sequences and amino acid substitutions are represented by color intensity (*Figure 4—figure supplements 1–3*). For each lineage, inferred intermediate sequences found on the most probable lineage paths were selected for study (*Supplementary file 1*, *Figure 4—figure supplements 1–3*).

In the 016-VL lineage, we determined that a 3-nt insertion event occurred prior to the inferred_7_712 intermediate within this antibody's most likely developmental pathway (*Figure 4—figure supplement 1*). Since most insertion-containing clonal NGS sequences (136 of 142) encoded serine at the insertion site (amino acid position 94), we inserted S94 into 016-2$_{VL}$ instead of N94 that would correspond to the mature 016-VL sequence. For thoroughness, the inferred_5_789 016-VL intermediate (which precedes inferred_7_712) was synthesized both without (016-1$_{VL}$) and with (016-2$_{VL}$) the S94 insertion.

## Identification of lineage-like NGS sequences

To determine if the computationally-inferred naïve and ancestor sequences were observed in the NGS data, we performed the following procedure for each lineage (implemented in a script found here: https://git.io/Je7Zp). A local BLAST database was created for each seeded clonal family and queried for sampled sequences that had high nucleotide sequence identity to lineage members using the 'blastn' command (Biopython package). E-value of 0.001 was used; other settings were default. Blastn matches for each lineage member were sorted according to their percent nt identity and alignment length. Sampled sequences with 100% nt or 100% aa identity in common with lineage members were noted (*Supplementary file 1*, *Figure 4—figure supplements 1–3*). To identify the closest sampled sequences to each VH mature sequence, blastn results per VH mature query were viewed and the highest percent aa identity match was selected.

## ADCC-focused lineage determination

As outlined in our Results, we first selected 'middle' lineage intermediates for studying ADCC gain-of-function based on their moderate inclusion of amino acid substitutions (between naïve and mature sequences; *Supplementary file 1*) and high statistical confidence (*Figure 4—figure supplements 1–3*), along with, in some cases, their validation by sampled NGS sequences (*Figure 4—figure supplements 1–3*), and/or concentration of substitutions in complementarity-determining regions (CDRs) (*Supplementary file 1*). Middle intermediate chains were paired with partner naïve, middle, and mature chains and tested for antigen binding and ADCC function. Based on preliminary results using middle intermediates, ADCC-focused lineages were chosen from remaining pre-middle inferred intermediates or post-middle intermediates depending on whether the middle intermediate chain contributed to ADCC activity. If multiple relevant intermediate choices were available within the pertinent (early or late) portion of an inferred lineage (*Figure 4—figure supplements 1–3*), we selected sequences that had amino acid substitutions that were concentrated in CDRs (*Supplementary file 1*) and, whenever possible, we selected intermediates that were validated by NGS-sampled sequences, as aforementioned (*Figure 4—figure supplements 1–3*). For 067-VH, 067-VK, 072-VH, and 105-VH lineages, early mutations were implicated in ADCC gain-of-function, but we lacked early lineage resolution and therefore could not select early inferred intermediates to study. Instead, we selectively incorporated CDR substitutions from the middle intermediate sequence into the inferred naïve sequence. Non-conservative FWR3 substitutions were also incorporated into the 067-VH early lineage. For each lineage, intermediate sequences were numbered consecutively based on their position in the developmental pathway between naïve (0) and mature (m). Heavy and light chain pairings for lineage intermediates were based on chronology, which reflected levels of mutation. Each chain was paired with several partner chains for functional assessment.

Ultimately, developmental lineages featuring the minimal number of mAbs to study ADCC development were defined based on levels of mAb mutation and ADCC activity. For each lineage, the minimal antibodies included (1) the inferred naïve, (2) the most mutated intermediate(s) that lacked ADCC activity, if available, (3) the least mutated intermediate(s) that gained detectable ADCC

function, and (4) the least mutated intermediate(s) that demonstrated ADCC activity comparable to the mature antibodies (>50%). Mature antibodies were always included as benchmark positive controls.

## Cell lines

For antibody production: HEK 293 F cells (RRID:CVCL_D603; originally derived from female human embryonic kidney cells) were obtained from Invitrogen (Thermo Fisher Scientific, Waltham, MA, catalog #R790-07) and grown at 37°C in Freestyle 293 Expression Medium (Thermo Fisher Scientific, catalog #12338002) in baffle-bottomed flasks orbiting at 135 rpm. These cells were not further authenticated in our hands.

For RFADCC: CEM.NKR cells (RRID:CVCL_X622; originally derived from female human T-lymphoblastoid cells) were obtained from NIH AIDS Reagent Program (Germantown, MD, catalog #458) and grown at 37°C in RPMI 1640 media with added penicillin (100 U/mL), streptomycin (100 µg/mL), Amphotericin B (250 ng/mL), L-glutamine (2 mM), and fetal bovine serum (10%) (all from Thermo Fisher Scientific). These cells were not further authenticated in our hands.

## Monoclonal antibody production

Antibody heavy and lightchain variable regions were synthesized as FragmentGENES (GENEWIZ, South Plainfield, NJ) and subsequently cloned into corresponding Igγ1, Igκ or Igλ expression vectors (*Tiller et al., 2008*). Equal ratios of heavy and light chain plasmids were co-transfected into HEK 293 F cells using FreeStyle MAX (Thermo Fisher Scientific, catalog #16447100) according to the manufacturer's instructions. Column-based Pierce protein G (Thermo Fisher Scientific, catalog #20397) purification of IgG was done according to the manufacturer's instructions.

## Rapid and fluorometric ADCC (RFADCC) assay

The RFADCC assay was performed as described (*Gómez-Román et al., 2006*; *Williams et al., 2019*). In short, CEM-NKr cells (NIH AIDS Reagent Program, catalog #458) were double-labeled with PKH-26-cell membrane dye (Sigma-Aldrich, St. Louis, MO) and a cytoplasmic-staining dye (Vybrant CFDA SE Cell Tracer Kit, Thermo Fisher Scientific). The double-labeled cells were coated with clade A gp120 (BL035.W6M.Env.C1, *Wu et al., 2006*) or clade C gp41 ectodomain (C.ZA.1197MB) (*Rousseau et al., 2006*) for 1 hr at room temperature at a ratio of 1.5 µg protein: $1 \times 10^5$ double-stained target cells. For RFADCC breadth assays, additional HIV gp120 monomers of the following strains were: QA255.22P.A17, BG505.W6M.B1, CAP210.2.00.E8, MK184.W0M.G3, and SF162. All gp120 and gp41 proteins were sourced from Immune Technology Corp, New York, NY, or Cambridge Biologics, Brookline, MA, as specified in the Key Resources Table. Coated targets were washed once with complete RMPI media (RPMI supplemented with 10% FBS, 4.0 mM Glutamax, and 1% antibiotic-antimycotic, all from Thermo Fisher Scientific). Monoclonal antibodies were diluted in complete RPMI media to a concentration of 100–500 ng/mL and mixed with $5 \times 10^3$ coated target cells for 10 min at room temperature. PBMCs (peripheral blood mononuclear cells; Bloodworks Northwest, Seattle, WA) from an HIV-negative donor were added at a ratio of 50 effector cells: one target cell. The coated target cells, antibodies, and effector cells were co-cultured for 4 hr at 37°C then fixed in 1% paraformaldehyde (Affymetrix, Santa Clara, CA). Cells were analyzed by flow cytometry (Symphony I/II, BD Biosciences, San Jose, CA) and ADCC activity was defined as the percent of PKH-26+ CFDA- cells after background subtraction, where background (antibody-mediated killing of uncoated cells) was standardized to be 3–5% as analyzed using FlowJo software (FlowJo LLC, Ashland, OR). To mitigate differences in activity observed with different PBMC donors and between experiments, ADCC activity for each sample was normalized to monoclonal positive control mAbs: 167-D (NIH AIDS Reagent Program, Cat #11681) for gp41-targeted Abs and C11 (NIH AIDS Reagent Program, Cat #7374) for gp120-targeted Abs. The activity of an unrelated antibody, FI6v3, that recognizes influenza hemagglutinin protein was used to define the limit of detection. For each replicate experiment, samples were categorized in the following manner: positive (sample >2*FI6v3), indeterminate (1*FI6v3 ≤ sample ≤2*FI6v3), negative (sample <1*FI6v3). Experiments were excluded if FI6v3 signal was >10% of monoclonal positive control signal. Background (uncoated cells) and negative-control (1*FI6v3) signal was subtracted from each sample's activity and the resultant values were averaged across experimental replicates, normalized to the respective lineage's mature mAb

activity, and plotted in Prism v8.0c (GraphPad, San Diego, CA). A designation of indeterminate was also assigned in cases where samples were negative in the majority of experimental replicates, but indeterminate or positive in any replicate(s). In such cases, average activity was reported as usual.

## Biolayer interferometry

QA255 monoclonal antibody binding to monomeric gp120 or gp41 was measured using biolayer interferometry on an Octet RED instrument (ForteBio, Fremont, CA). Antibodies diluted to 8 µg mL$^{-1}$ in a filtered buffer solution of 1X PBS containing 0.1% BSA, 0.005% Tween-20, and 0.02% sodium azide were immobilized onto anti-human IgG Fc capture biosensors (ForteBio). C. ZA.1197MB or BL035.W6M.C1 gp41 was diluted to 250 nM, or as indicated, in the same buffer (above) and a series of up to six, two-fold dilutions were tested as analytes in solution at 30°C. The kinetics of mAb binding were measured as follows: association was monitored for 180 s, dissociation was monitored for 180 s, and regeneration was performed in 10 mM Glycine HCl (pH 1.5). For experiments in which dilution series were run, binding-affinity constants ($K_D$; on-rate, $K_{on}$; off-rate, $K_{dis}$) were calculated using ForteBio's Data Analysis Software 7.0. Responses (nanometer shift) were calculated and background-subtracted using double referencing against the buffer reference signal and non-specific binding of biosensor to analyte. Data were processed by Savitzky-Golay filtering prior to fitting using a 1:1 model of binding kinetics.

## Epitope mapping with phage-DMS

Profiling of escape mutations was done as previously described (*Garrett et al., 2020*). We utilized a deep mutational scanning (DMS) phage display library containing wildtype and mutant peptides that tile across the gp41 and V3 portions of three Envelope strains (BG505.W6M.C2, BF520.W14M.C2, and C.ZA.1197MB). Peptides in the library are 31 amino acids long and contain every possible single amino acid mutation at the central position, with peptides overlapping by 30 amino acids. Immuno-precipitation was performed in technical duplicate by incubating 10 ng of antibody sample with 1 mL of gp41/V3 Phage-DMS library at a concentration representing 200,000 pfu/mL of each unique clone. Antibody and phage were allowed to form complexes by rocking overnight at 4°C in 96-deep-well plates that had been pre-blocked with 3% BSA in TBST. To isolate phage-antibody complexes, 20 uL each of Protein A and Protein G Dynabeads were added and incubated at 4°C for 4 hr. Beads were magnetically separated, washed 3x with 400 µL wash buffer (150 mM NaCl, 50 mM Tris-HCl, 0.1% [vol/vol] NP-40, pH 7.5), resuspended in 40 µL of dH2O before lysis (95°C for 10 min), and stored at −20°C. PCR and deep sequencing of the enriched phage was done as previously described. Briefly, sequences were amplified to add Illumina barcodes (two rounds) and then pooled and sequenced on an Illumina MiSeq with 1 × 125 bp single-end reads (BioProject accession PRJNA685289). Enrichment and scaled differential selection were calculated as previously described, with analysis performed and plots generated using RStudio (*Figure 9—source code 1*, *Figure 9—source datas 1–3*). Two biological replicates were run, using the same antibody preps but distinct peptide library preps.

## Data and code availability

The QA255 longitudinal antibody deep sequencing datasets (*Table 1*) and Phage-DMS sequencing data sets generated during this study are publicly available at BioProject SRA, accessions PRJNA639297 [https://www.ncbi.nlm.nih.gov/bioproject/PRJNA639297/] and PRJNA685289 [https://www.ncbi.nlm.nih.gov/bioproject/PRJNA685289]. The inferred antibody variable region sequences generated in this study have not been deposited in GenBank because computationally-inferred sequences are not accepted, but they are available in *Supplementary file 1*. GenBank accession numbers for mature QA255 antibody are MT791224-MT791235 and QA255 Envelope sequences are MW383929-MW383956. GenBank accession numbers for HIV Env variants are available in the Key Resources Table. The custom code generated or used in this study for antibody lineage determination is publicly available on GitHub: prune.py (https://github.com/matsengrp/cft/blob/master/bin/prune.py), ecgtheow (https://github.com/matsengrp/ecgtheow), CFT (https://github.com/matsengrp/cft), and Blast validation (https://git.io/Je7Zp). Code for Phage-DMS analysis is provided as *Figure 9—source code 1*.

## Quantification and statistical analysis

Where applicable, raw data were normalized and/or averaged across replicates using Microsoft Excel. Plots were generated using GraphPad Prism version 8.0 c. Relevant experimental details, such as use of biological and technical replicates, can be found in figure legends. For RFADCC plots, bars with error bars represent mean and SD. RFADCC experimental exclusion criteria are detailed within the RFADCC method section above. For Phage-DMS, calculations were performed as previously described (*Garrett et al., 2020*), using RStudio; Phage-DMS analysis source code is available in *Figure 9—source code 1*.

## Acknowledgements

We thank the participants, staff, Scott McClelland, and Ludo Lavreys for continued efforts to oversee the Mombasa cohort. We acknowledge Brian Oliver, Vladimir Vigdorovich, and D Noah Sather for technical assistance with NGS library preparation and sequence processing. We thank Vrasha Chohan, Haidyn Weight, and Tucker Price for their help with antibody production and testing.

## Additional information

### Competing interests

Julie M Overbaugh: Reviewing editor, *eLife*. The other authors declare that no competing interests exist.

### Funding

| Funder | Grant reference number | Author |
| --- | --- | --- |
| National Institutes of Health | R37 AI038518 | Julie M Overbaugh |
| National Institutes of Health | R01 HD103571 | Julie M Overbaugh |
| National Institutes of Health | R01 GM113246 | Frederick A Matsen IV |
| National Institutes of Health | R01 AI146028 | Frederick A Matsen IV |
| National Institutes of Health | T32 AI07140 | Laura E Doepker |
| National Institutes of Health | T32 AI083203 | Zak Yaffe |
| National Institutes of Health | P30 AI027757 | Duncan K Ralph |
| Howard Hughes Medical Institute | Faculty Scholar grant | Frederick A Matsen IV |
| Simons Foundation | Faculty Scholar grant | Frederick A Matsen IV |

The funders had no role in study design, data collection and interpretation, or the decision to submit the work for publication.

### Author contributions

Laura E Doepker, Conceptualization, Data curation, Formal analysis, Supervision, Validation, Investigation, Visualization, Methodology, Writing - original draft, Project administration, Writing - review and editing; Sonja Danon, Validation, Investigation, Visualization, Writing - original draft, Project administration, Writing - review and editing; Elias Harkins, Data curation, Software, Formal analysis, Validation, Visualization, Methodology, Writing - review and editing; Duncan K Ralph, Data curation, Software, Formal analysis, Validation, Methodology, Writing - review and editing; Zak Yaffe, Dana Arenz, Validation, Investigation, Writing - review and editing; Meghan E Garrett, Software, Validation, Investigation, Visualization, Methodology, Writing - original draft; Amrit Dhar, Data curation, Software, Formal analysis, Validation, Writing - review and editing; Cassia Wagner, Formal analysis, Validation, Methodology, Writing - review and editing; Megan M Stumpf, Formal analysis, Investigation, Methodology, Writing - review and editing; James A Williams, Supervision, Validation, Methodology, Writing - review and editing; Walter Jaoko, Kishor Mandaliya, Resources, Data curation; Kelly K Lee, Resources, Supervision, Methodology, Writing - review and editing; Frederick A Matsen IV,

Resources, Software, Supervision, Writing - review and editing; Julie M Overbaugh, Conceptualization, Resources, Supervision, Funding acquisition, Writing - original draft, Project administration, Writing - review and editing

### Author ORCIDs
Laura E Doepker ⬥ https://orcid.org/0000-0003-4514-5003
Sonja Danon ⬥ http://orcid.org/0000-0002-5399-7081
Cassia Wagner ⬥ http://orcid.org/0000-0002-9934-7578
Megan M Stumpf ⬥ http://orcid.org/0000-0001-8085-3094
Frederick A Matsen IV ⬥ http://orcid.org/0000-0003-0607-6025
Julie M Overbaugh ⬥ https://orcid.org/0000-0002-0239-9444

### Ethics
Human subjects: Approval to conduct this study was provided by the ethical review committees of the University of Nairobi Institutional Review Board, the Fred Hutchinson Cancer Research Center Institutional Review Board (protocol 7776), and the University of Washington Institutional Review Board; Clinical Trial Management System Number RG1000880. Study participants provided written informed consent prior to enrollment.

### Decision letter and Author response
Decision letter https://doi.org/10.7554/eLife.63444.sa1
Author response https://doi.org/10.7554/eLife.63444.sa2

## Additional files

### Supplementary files
• Supplementary file 1. Sequences of six ADCC antibody lineages. (fasta file) Included are inferred naïve, computationally-inferred lineage intermediate, manually- inferred lineage intermediate, corrected mature, and NGS sequences with high sequence identity to lineage members and/or mature sequences. Sequences are provided as, because computationally inferred sequences cannot be deposited into GenBank.

• Supplementary file 2. Select NGS sequences with highest sequence identity to mature D914 mAb sequences. (fasta file)

• Transparent reporting form

### Data availability
Sequencing data have been deposited in BioProject SRA under the accession codes PRJNA639297 and PRJNA685289. Data generated and analyzed in this study are included in the manuscript and supporting files. Source data files have been provided for Figures 1, 2, 4, 5, 7, and 8.

The following datasets were generated:

| Author(s) | Year | Dataset title | Dataset URL | Database and Identifier |
|---|---|---|---|---|
| Doepker L, Overbaugh J | 2020 | Subject QA255 antibody sequencing | https://www.ncbi.nlm.nih.gov/bioproject/PRJNA639297 | NCBI BioProject, PRJNA639297 |
| Garrett M, Overbaugh J | 2020 | Development of antibody-dependent cell cytotoxicity function in HIV-1 antibodies | https://www.ncbi.nlm.nih.gov/bioproject/PRJNA685289/ | NCBI BioProject, PRJNA685289 |

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
