## [Decision Letter]

**Acceptance summary:**

The manuscript uses a time series of bleeds from a single HIV-infected individual with low neutralising but high antibody-dependent cellular cytotoxicity (ADCC) antibody activity to identify six temporal lineages culminating in six distinct ADCC-capable antibodies. Using variable region repertoire sequencing, computational sequence reconstruction of antibody intermediates, and measurements of affinity and ADCC efficiency, the authors provide insights into how binding affinity and ADCC capability likely change during antibody clone diversification over time. Similar to studies that have investigated HIV-specific neutralising antibodies, this analysis provides clues to the evolutionary pathways of ADCC-functional antibody development.

**Decision letter after peer review:**

Thank you for submitting your article "Development of antibody-dependent cell cytotoxicity function in HIV-1 antibodies" for consideration by *eLife*. Your article has been reviewed by three peer reviewers, and the evaluation has been overseen by Satyajit Rath as the Senior and Reviewing Editor. The following individuals involved in review of your submission have agreed to reveal their identity: Stephen Kent (Reviewer #1); Felix Horns (Reviewer #3).

The reviewers have discussed the reviews with one another and the Reviewing Editor has drafted this decision to help you prepare a revised submission.

Summary:

The manuscript analyses the origins and development of six HIV-specific ADCC-mediating antibodies from a single HIV clade A-infected subject over time, beginning from seroconversion to 4 years post-infection. Using NGS-based variable region repertoire sequencing, computational sequence reconstruction of antibody intermediates, and measurements of affinity and ADCC efficiency, the authors provide insights into how binding affinity and ADCC capability likely change during antibody clone diversification over time. Similar to studies that have investigated HIV-specific neutralising antibodies, this analysis provides clues to the evolutionary pathways of ADCC-functional antibody development. While antibody binding was correlated with ADCC activity, antibodies with similar affinity exhibited ADCC activity with different potencies. Mutations in complementary determining regions were needed for ADCC activity, with additional mutations in framework regions generally providing enhanced potency. The study was rigorously conducted, and addresses an important question with well-supported conclusions. Some issues that need to be addressed are identified below.

Essential revisions:

1) The ordering of mutations is not fully resolved and thus some claims about ordering are not well supported. As an illustrative example, Figure 3A shows that several mutations were acquired between 0_VH_0_VL_ and 1_VH_0_VL_ that improved both ADCC and affinity. The ordering of these mutations and thus the ordering of improvements to ADCC and affinity are not resolved. Similarly, further mutations in 1_VH_1_VL_ dramatically improved both ADCC and affinity. But again the ordering is not resolved. Similar arguments can be made for most of the antibody clones in Figure 3 and Figure 4. Thus, it seems misleading to claim that mutations first improve affinity, then later mutations improve ADCC, as implied in the Abstract. Several places elsewhere in the text also imply that the ordering of mutations has been resolved (e.g. "Thereafter, six light chain mutations among CDRL1, CDRL3, and FWRL2 augmented gp41 binding and ADCC potency in 006-1_VH_1_VL_"). Is there any evidence supporting this ordering? In the absence of evidence, it seems quite likely that these mutations happened in an interlaced fashion. This should be clarified in the text.

2) The fidelity of pairing between heavy and light chains is a key issue and should be clarified. In this study, the heavy and light chain sequences are reconstructed separately based on the mature antibody sequences. The experimental results hinge significantly on correct identification of corresponding heavy and light chains. The results would be more easily interpreted and perhaps better supported if the authors could estimate the fidelity of this strategy of computational pairing, possibly using a "ground truth" from paired heavy-light chain sequencing data sets. Otherwise, the authors should acknowledge this caveat, ideally in the main text.

3) Again, because the authors have reconstructed the mutation histories of heavy and light chains separately, we have no direct information about the co-occurrence of heavy and light chain mutations. Thus, the pairing of heavy and light chain sequences and the mutations they carry is arbitrary and may not represent actual combinations of mutations that were present together in a true intermediate. Ideally, the authors would validate or estimate the fidelity of the relative timing or co-occurrence of the mutations. This could perhaps be achieved using a paired heavy-light chain sequencing data set, as above. Otherwise, the authors should clarify in the text that the reconstructed intermediates (e.g. 1_VH_ 1_VL_, 1_VH_ 2_VL_, and 2_VH_ 2_VL_ in Figure 3B) are arbitrarily paired and may not represent the combinations of mutations in true intermediates.

4) A major concern is the lack of analysis of the native antibody Fc region over time given the central nature of this part of the antibody to ADCC functions. It may that there are no changes or no insights that can be drawn from this. However that would be an important finding in of itself and show the importance of the Fab analyses. Such an analysis would make the manuscript much stronger.

5) The first time point sampled is quite late at over 400 days. Comments about putative changes earlier would be useful.

6) There is interest in potential escape from ADCC antibodies as an argument about the pressure applied by these responses – with their evolution studies and knowledge of the epitopes the authors may be able to draw inferences on ADCC escape, and comments regarding this issue would be useful.

7) It would strengthen the argument to include assessment of the breadth of ADCC activity and how it fits into the evolutionary pathways.

---

## [Author Response]

Essential revisions:1) The ordering of mutations is not fully resolved and thus some claims about ordering are not well supported. As an illustrative example, Figure 3A shows that several mutations were acquired between 0_VH_0_VL_ and 1_VH_0_VL_ that improved both ADCC and affinity. The ordering of these mutations and thus the ordering of improvements to ADCC and affinity are not resolved. Similarly, further mutations in 1_VH_1_VL_ dramatically improved both ADCC and affinity. But again the ordering is not resolved. Similar arguments can be made for most of the antibody clones in Figure 3 and Figure 4. Thus, it seems misleading to claim that mutations first improve affinity, then later mutations improve ADCC, as implied in the Abstract. Several places elsewhere in the text also imply that the ordering of mutations has been resolved (e.g. "Thereafter, six light chain mutations among CDRL1, CDRL3, and FWRL2 augmented gp41 binding and ADCC potency in 006-1_VH_1_VL_"). Is there any evidence supporting this ordering? In the absence of evidence, it seems quite likely that these mutations happened in an interlaced fashion. This should be clarified in the text.

To address points 1, 2, 3 and 5, that raise concerns about claims of developmental chronology and pairing choices amongst heavy and light chain intermediates, we’d like to offer clearer explanations of our approaches (here) as well as amendments to the text based (listed per point).

For longitudinal antibody sequencing, we prioritized depth rather than per-sequence accuracy because we already had “ground truth” pairing information from our six biologically-isolated ADCC antibodies and our focus was to infer likely ancestral sequences (nodes) between naïve and mature sequences within each clonal phylogeny. Our choices of sequencing and analysis methods were also driven by the characteristics of our unique, precious, and old samples: they derive from the pre-ART time period, which is important but means they have been in -80°C storage for ~20 years. With more abundant samples, we would have been eager to perform multiple sequencing approaches, such as both bulk unpaired and paired sequencing, but that was not the case here.

To achieve the highest depth possible with our longitudinal samples, we performed unpaired bulk sequencing with a well-established protocol rather than paired sequencing, which severely curtails depth and was not a well-practiced approach at the time this study began. As described and validated in a previous study of ours (Simonich et al., 2019), we designed our analysis to handle the challenges coming from sparsely-sampled noisy data. Specifically, we used Bayesian phylogenetics to infer ancestral sequences while allowing for uncertainty in the tree, rather than using ancestral sequence reconstructions from a single maximum-likelihood tree, which does not acknowledge tree uncertainty that can be quite significant. The chronology (ordering) of these ancestral sequences is resolved as clearly as the sequencing data and analyses allow; some pathways are resolved to 1-2 AA substitutions per ordered step, while others lack fine resolution and big leaps containing many AA substitutions occur between inferred ancestral sequences. The resolution of each lineage is viewable in Figure 4—figure supplements 1-3, with a main text example now included as Figure 4.

A strength of our approach is that we developed specific tools to enable confident reconstruction of the ordering of ancestral sequences (and, thus, the mutations occurring between them) even though our biological samples are quite old. In other words, we did not use off-the-shelf computational methods developed for other biological problems and adapt them for antibody sequence studies. This is exciting, indeed; if we had had more samples, we wouldn’t have had to work so hard to this end. All in all, with the bulk longitudinal sequencing data we acquired, we made as many confident conclusions about which mutations came before others as we could and those intermediate sequences are what we paired together into mAbs and tested.

Since this approach inferred the most likely development routes taken by individual antibody chains (heavy and light, independently), our inferred ancestral sequences are indeed unpaired, as emphasized by the reviewers’ points 1, 2, and 3. As outlined below, we added text to more explicitly explain that our experimental pairings do not reflect true biological intermediates. Indeed, we did not set out to define with certainty the paired intermediates that led to antibody ADCC function. Rather, our goal was to define biologically-informed minimal mutations (ideally in the form of a highly-likely computationally inferred ancestral sequence) that are needed for antibody lineages to gain ADCC function. Ultimately, we used the sequencing data and computational analysis to learn what is probable in terms of antibody development over time and, in the end, the findings from this approach are what drive this story – namely that both CDR and FWR mutations are needed to achieve potent ADCC and binding affinity is not exclusively responsible for ADCC potency. Moreover, this is the first study of stepwise ADCC antibody development ever, and we find it frankly remarkable that we found themes among the six lineages under investigation.

Changes to the text in response to points 1, 2, and 3:

– Clarified explanations of our methods which determined the chronology of highly probable ancestral sequences within single chains (heavy or light), though not each individual mutation per chain (Introduction, Results, Discussion and Materials and methods).

– Amended text that was inaccurately implying chronology between heavy and light chain mutations (Abstract, Results and Discussion).

2) The fidelity of pairing between heavy and light chains is a key issue and should be clarified. In this study, the heavy and light chain sequences are reconstructed separately based on the mature antibody sequences. The experimental results hinge significantly on correct identification of corresponding heavy and light chains. The results would be more easily interpreted and perhaps better supported if the authors could estimate the fidelity of this strategy of computational pairing, possibly using a "ground truth" from paired heavy-light chain sequencing data sets. Otherwise, the authors should acknowledge this caveat, ideally in the main text.

These concerns are addressed above in response to points 1, 2, 3, and 5, including the list of changes to the text.

3) Again, because the authors have reconstructed the mutation histories of heavy and light chains separately, we have no direct information about the co-occurrence of heavy and light chain mutations. Thus, the pairing of heavy and light chain sequences and the mutations they carry is arbitrary and may not represent actual combinations of mutations that were present together in a true intermediate. Ideally, the authors would validate or estimate the fidelity of the relative timing or co-occurrence of the mutations. This could perhaps be achieved using a paired heavy-light chain sequencing data set, as above. Otherwise, the authors should clarify in the text that the reconstructed intermediates (e.g. 1_VH_ 1_VL_, 1_VH_ 2_VL_, and 2_VH_ 2_VL_ in Figure 3B) are arbitrarily paired and may not represent the combinations of mutations in true intermediates.

These concerns are addressed above in response to points 1, 2, 3, and 5, including the list of changes to the text.

4) A major concern is the lack of analysis of the native antibody Fc region over time given the central nature of this part of the antibody to ADCC functions. It may that there are no changes or no insights that can be drawn from this. However that would be an important finding in of itself and show the importance of the Fab analyses. Such an analysis would make the manuscript much stronger.

Based on literature in this field, the Fc region of antibodies (actually the whole constant region, part of which is included in the Fab portion), remains stable in human antibodies. Specifically, the constant region in antibodies lacks mutations because the AID (activation-induced cytidine deaminase) enzyme responsible for somatic hypermutation does not gain access to the constant regions of Ig genes (Longerich et al., 2005; Peled et al., 2008) even though AID is required for class switch recombination, which takes place upstream of the 5’ ends of each constant heavy chain gene (Muramatsu et al., 2000; Stavnezer, Guikema and Schrader, 2008). With these notions in mind, we chose to perform full length variable region sequencing, which included enough sequence into the constant region to determine antibody isotype. This approach allowed us to determine the effect of AID-mediated somatic hypermutation on ADCC development in antibody lineages and, indeed, our data demonstrate that variable region mutations affect ADCC function even though ADCC activity is an Fc-mediated process.

However, we hear and completely agree with your point that ADCC activity is Fc-mediated and information about the Fc portions of these antibodies is of interest (sequence, binding angle, isotype and subclass). We have clarified that all these antibodies are IgG1 subclass in the Introduction. Despite the abovementioned dogma about constant regions remaining unmutated through antibody lineage evolution, we regret that we lack enough samples to fully sequence the constant regions of these lineages, because, if we found mutations in these regions, it would indeed be an important and extremely surprising and novel finding.

5) The first time point sampled is quite late at over 400 days. Comments about putative changes earlier would be useful.

In the above response to points 1, 2, 3, and 5, we clarify the extent of our abilities to predict ordered changes based on clonal antibody sequences from various time points. In response to this comment, specifically, we have added time point information to the NGS hits (green and yellow stars) within pathways presented in Figure 4—figure supplements 1-3. Thus, readers may infer which inferred ancestral sequences were possibly present at/after distinct time points post-infection. Though we cannot confidently dissect the order of mutations that occurred earlier than day 462 post-infection any further than we already have, we have added text in the Discussion to note that three VH clonal families (006, 067, and 105) contain only D462 NGS sequences (Table 2) and thus all developmental inferences for these three families likely occurred prior to D462 (Discussion).

6) There is interest in potential escape from ADCC antibodies as an argument about the pressure applied by these responses – with their evolution studies and knowledge of the epitopes the authors may be able to draw inferences on ADCC escape, and comments regarding this issue would be useful.

We have added data to the Results section on the (lack of) evidence for longitudinal mutational Env escape from lineages 067, 072, and 105, which all have linear peptide epitopes. The 067 and 072 epitopes are entirely stable between 21 dpi and 1729 dpi. For lineage 105, we finely mapped the epitope of both the inferred naïve mAb and the mature mAb, finding that the residue positions comprising the epitope remain stable, and that there is little evidence for strong selective pressure on these residues between 21 dpi and 1729 dpi. In adding the additional section on 105 lineage epitope mapping, we have also added Meghan Garrett as an author of the paper for her contributions to this work. Unfortunately, lineages 006, 016, and 157 target discontinuous epitopes, so we cannot comment on escape here without significant additional experimentation.

Changes to the text: Results, Discussion, Materials and methods, Figure 9, Figure 9—figure supplements 1-2, Figure 9—source data 1, Figure 9—source data 1-3.

7) It would strengthen the argument to include assessment of the breadth of ADCC activity and how it fits into the evolutionary pathways.

We have added an assessment of developing ADCC breadth in the two gp120-targeted lineages as possible, given their mature breadth profiles (Williams et al., 2015). Specifically, we added data for RFADCC against two (lineage 105) or five (lineage 157) additional HIV strains, including the autologous transmitted founder virus (lineage 157 only). Changes to the text include the addition of Figure 6—figure supplement 1 and in the Results, Discussion and Materials and methods. The results indicate that full breadth developed after initial ADCC activity by both lineages. These data indeed strengthen the study; thank you for the suggestion. We maintain confidence in BL035 gp120 as the primary RFADCC coat protein for gp120-targeted antibodies in this study because we previously demonstrated that this strain is representative of RFADCC results among seven different gp120s of various clades (Milligan et al., 2015). This experimental choice is now clarified in the Discussion.

Unfortunately, RFADCC breadth studies are not possible for gp41-targeted lineages. There are very limited gp41 coat proteins available for experimental use and early attempts with MN gp41 yielded no RFADCC signal. As of yet, C.ZA.1197MB gp41 ectodomain is the only effective RFADCC coat protein available for gp41-targeted lineages.

References:

Longerich, S., Tanaka, A., Bozek, G., Nicolae, D., and Storb, U. (2005). The very 5' end and the constant region of Ig genes are spared from somatic mutation because AID does not access these regions. J Exp Med, 202(10), 1443-1454. doi:10.1084/jem.20051604Muramatsu, M., Kinoshita, K., Fagarasan, S., Yamada, S., Shinkai, Y., and Honjo, T. (2000). Class switch recombination and hypermutation require activation-induced cytidine deaminase (AID), a potential RNA editing enzyme. Cell, 102(5), 553-563. doi:10.1016/s0092-8674(00)00078-7Peled, J. U., Kuang, F. L., Iglesias-Ussel, M. D., Roa, S., Kalis, S. L., Goodman, M. F., and Scharff, M. D. (2008). The biochemistry of somatic hypermutation. Annu Rev Immunol, 26, 481-511. doi:10.1146/annurev.immunol.26.021607.090236Stavnezer, J., Guikema, J. E., and Schrader, C. E. (2008). Mechanism and regulation of class switch recombination. Annu Rev Immunol, 26, 261-292. doi:10.1146/annurev.immunol.26.021607.090248